# Model-Based Reparameterization Policy Gradient Methods: Theory and Practical Algorithms

**Shenao Zhang**[1]     **Boyi Liu**[1]     **Zhaoran Wang**[1†]     **Tuo Zhao**[2†]

[1]Northwestern University    [2]Georgia Tech

shenao@u.northwestern.edu

## Abstract

ReParameterization (RP) Policy Gradient Methods (PGMs) have been widely adopted for continuous control tasks in robotics and computer graphics. However, recent studies have revealed that, when applied to long-term reinforcement learning problems, model-based RP PGMs may experience chaotic and non-smooth optimization landscapes with exploding gradient variance, which leads to slow convergence. This is in contrast to the conventional belief that reparameterization methods have low gradient estimation variance in problems such as training deep generative models. To comprehend this phenomenon, we conduct a theoretical examination of model-based RP PGMs and search for solutions to the optimization difficulties. Specifically, we analyze the convergence of the model-based RP PGMs and pinpoint the smoothness of function approximators as a major factor that affects the quality of gradient estimation. Based on our analysis, we propose a spectral normalization method to mitigate the exploding variance issue caused by long model unrolls. Our experimental results demonstrate that proper normalization significantly reduces the gradient variance of model-based RP PGMs. As a result, the performance of the proposed method is comparable or superior to other gradient estimators, such as the Likelihood Ratio (LR) gradient estimator. Our code is available at https://github.com/agentification/RP_PGM.

## 1 Introduction

Reinforcement Learning (RL) has seen tremendous success in a variety of sequential decision-making applications, such as strategy games [51, 59] and robotics [15, 61], by identifying actions that maximize long-term accumulated rewards. As one of the most popular methodologies, the policy gradient methods (PGM) [56, 31, 52] seek to search for the optimal policy by iteratively computing and following a stochastic gradient direction with respect to the policy parameters. Therefore, the quality of the stochastic gradient estimation is essential for the effectiveness of PGMs.

Two main categories have emerged in the realm of stochastic gradient estimation: (1) Likelihood Ratio (LR) estimators, which perform zeroth-order estimation through the sampling of function evaluations [64, 32, 31], and (2) ReParameterization (RP) gradient estimators, which harness the differentiability of the function approximation [17, 48, 12, 53]. Despite the wide adoption of both LR and RP PGMs in practice, the majority of the literature on the theoretical properties of PGMs focuses on LR PGMs. The optimality and approximation error of LR PGMs have been heavily investigated under various settings [3, 60, 7]. Conversely, the theoretical underpinnings of RP PGMs remain to be fully explored, with a dearth of research on the quality of RP gradient estimators and the convergence of RP PGMs.

RP gradient estimators have established themselves as a reliable technique for training deep generative models such as variational autoencoders [17]. From a stochastic optimization perspective, previous

---

† Equal advising.

37th Conference on Neural Information Processing Systems (NeurIPS 2023).

studies [48, 40] have shown that RP gradient methods enjoy small variance, which leads to better convergence and performance. However, recent research [42, 37] has reported an opposite observation: When applied to long-horizon reinforcement learning problems, model-based RP PGMs tend to encounter chaotic optimization procedures and highly non-smooth optimization landscapes with exploding gradient variance, causing slow convergence.

Such an intriguing phenomenon inspires us to delve deeper into the theoretical properties of RP gradient estimators in search of a remedy for the issue of exploding gradient variance in model-based RP PGMs. To this end, we present a unified theoretical framework for the examination of model-based RP PGMs and establish their convergence results. Our analysis implies that the smoothness and accuracy of the learned model are crucial determinants of the exploding variance of RP gradients: (1) both the gradient variance and bias exhibit a polynomial dependence on the Lipschitz continuity of the learned model and policy w.r.t. the input state, with degrees that increase linearly with the steps of model value expansion, and (2) the bias also depends on the error of the estimated model and value.

Our findings suggest that imposing smoothness on the model and policy can greatly decrease the variance of RP gradient estimators. To put this discovery into practice, we propose a spectral normalization method to enforce the smoothness of the learned model and policy. It's worth noting that this method can enhance the algorithm's efficiency without substantially compromising accuracy when the underlying transition kernel is smooth. However, if the transition kernel is not smooth, enforcing smoothness may lead to increased error in the learned model and introduce bias. In such cases, a balance should be struck between model bias and gradient variance. Nonetheless, our empirical study demonstrates that the reduced gradient variance when applying spectral normalization leads to a significant performance boost, even with the cost of a higher bias. Furthermore, our results highlight the potential of investigating model-based RP PGMs, as they demonstrate superiority over other model-based and Likelihood Ratio (LR) gradient estimator alternatives.

## 2 Background

**Reinforcement Learning.** We consider learning to optimize an infinite-horizon $\gamma$-discounted Markov Decision Process (MDP) over repeated episodes of interaction. We denote by $\mathcal{S} \subseteq \mathbb{R}^{d_s}$ and $\mathcal{A} \subseteq \mathbb{R}^{d_a}$ the state and action space, respectively. When taking an action $a \in \mathcal{A}$ at a state $s \in \mathcal{S}$, the agent receives a reward $r(s, a)$ and the MDP transits to a new state $s'$ according to $s' \sim f(\cdot \,|\, s, a)$.

We aim to find a policy $\pi$ that maps a state to an action distribution to maximize the expected cumulative reward. We denote by $V^\pi : \mathcal{S} \to \mathbb{R}$ and $Q^\pi : \mathcal{S} \times \mathcal{A} \to \mathbb{R}$ the state value function and the state-action value function associated with $\pi$, respectively, which are defined as follows,

$$Q^\pi(s, a) = (1 - \gamma) \cdot \mathbb{E}_{\pi, f}\left[\sum_{i=0}^\infty \gamma^i \cdot r(s_i, a_i) \,\middle|\, s_0 = s, a_0 = a\right], \quad V^\pi(s) = \mathbb{E}_{a \sim \pi}\left[Q^\pi(s, a)\right].$$

Here $s \in \mathcal{S}$, $a \in \mathcal{A}$, and the expectation $\mathbb{E}_{\pi, f}[\,\cdot\,]$ is taken with respect to the dynamic induced by the policy $\pi$ and the transition probability $f$. We denote by $\zeta$ the initial state distribution. Under policy $\pi$, the state and state-action visitation measure $\nu_\pi(s)$ over $\mathcal{S}$ and $\sigma_\pi(s, a)$ over $\mathcal{S} \times \mathcal{A}$ are defined as

$$\nu_\pi(s) = (1 - \gamma) \cdot \sum_{i=0}^\infty \gamma^i \cdot \mathbb{P}(s_i = s), \qquad \sigma_\pi(s, a) = (1 - \gamma) \cdot \sum_{i=0}^\infty \gamma^i \cdot \mathbb{P}(s_i = s, a_i = a),$$

where the summations are taken with respect to the trajectory induced by $s_0 \sim \zeta$, $a_i \sim \pi(\cdot \,|\, s_i)$, and $s_{i+1} \sim f(\cdot \,|\, s_i, a_i)$. The objective $J(\pi)$ of RL is defined as the expected policy value as follows,

$$J(\pi) = \mathbb{E}_{s_0 \sim \zeta}\left[V^\pi(s_0)\right] = \mathbb{E}_{(s,a) \sim \sigma_\pi}\left[r(s, a)\right]. \tag{2.1}$$

**Stochastic Gradient Estimation.** The underlying problem of policy gradient, i.e., computing the gradient of an expectation with respect to the parameters of the sampling distribution, takes the form $\nabla_\theta \mathbb{E}_{p(x;\theta)}[y(x)]$. To restore the RL objective, we can set $p(x; \theta)$ as the trajectory distribution conditioned on the policy parameter $\theta$ and $y(x)$ as the cumulative reward. In the sequel, we introduce two commonly used gradient estimators.

Likelihood Ratio (LR) Gradient (Zeroth-Order): By leveraging the *score function*, LR gradients only require samples of the function values. Since $\nabla_\theta \log p(x; \theta) = \nabla_\theta p(x; \theta)/p(x; \theta)$, the LR gradient is

$$\nabla_\theta \mathbb{E}_{p(x;\theta)}\left[y(x)\right] = \mathbb{E}_{p(x;\theta)}\left[y(x)\nabla_\theta \log p(x; \theta)\right]. \tag{2.2}$$

ReParameterization (RP) Gradient (First-Order): RP gradient benefits from the structural characteristics of the objective, i.e., how the overall objective is affected by the operations applied to the sources of randomness as they pass through the measure and into the cost function [40]. From the simulation property of continuous distribution, we have the following equivalent sampling processes:

$$\widehat{x} \sim p(x; \theta) \iff \widehat{x} = g(\widehat{\epsilon}, \theta), \widehat{\epsilon} \sim p(\epsilon), \tag{2.3}$$

which states that an alternative way to generate a sample $\widehat{x}$ from the distribution $p(x; \theta)$ is to sample from a simpler base distribution $p(\epsilon)$ and transform this sample using a deterministic function $g(\epsilon, \theta)$. Derived from the *law of the unconscious statistician* (LOTUS) [21], i.e., $\mathbb{E}_{p(x;\theta)}[y(x)] = \mathbb{E}_{p(\epsilon)}[y(g(\epsilon; \theta))]$, the RP gradient can be formulated as $\nabla_\theta \mathbb{E}_{p(x;\theta)}[y(x)] = \mathbb{E}_{p(\epsilon)}[\nabla_\theta y(g(\epsilon; \theta))]$.

# 3 Analytic Reparameterization Gradient in Reinforcement Learning

In this section, we present two fundamental *analytic* forms of the RP gradient in RL. We first consider the Policy-Value Gradient (PVG) method, which is model-free and can be expanded sequentially to obtain the Analytic Policy Gradient (APG) method. Then we discuss potential obstacles that may arise when developing practical algorithms.

We consider a policy $\pi_\theta(s, \varsigma)$ with noise $\varsigma$ in continuous action spaces. To ensure that the first-order gradient through the value function is well-defined, we make the following continuity assumption.

**Assumption 3.1** (Continuous MDP). We assume that $f(s' \mid s, a)$, $\pi_\theta(s, \varsigma)$[1], $r(s, a)$, and $\nabla_a r(s, a)$ are continuous in all parameters and variables $s$, $a$, $s'$.

**Policy-Value Gradient.** The reparameterization PVG takes the following general form,

$$\nabla_\theta J(\pi_\theta) = \mathbb{E}_{s \sim \zeta, \varsigma \sim p} \Big[ \nabla_\theta Q^{\pi_\theta}\big(s, \pi_\theta(s, \varsigma)\big) \Big]. \tag{3.1}$$

In sequential decision-making, any immediate action could lead to changes in all future states and rewards. Therefore, the value gradient $\nabla_\theta Q^{\pi_\theta}$ possesses a recursive structure. Adapted from the deterministic policy gradient theorem [52, 34] by considering stochasticity, we rewrite (3.1) as

$$\nabla_\theta J(\pi_\theta) = \mathbb{E}_{s \sim \nu_\pi, \varsigma} \Big[ \nabla_\theta \pi_\theta(s, \varsigma) \cdot \nabla_a Q^{\pi_\theta}(s, a) \big|_{a = \pi_\theta(s, \varsigma)} \Big].$$

Here, $\nabla_a Q^{\pi_\theta}$ can be estimated using a critic, which leads to model-free frameworks [25, 4]. Notably, as a result of the recursive structure of $\nabla_\theta Q^{\pi_\theta}$, the expectation is taken over the state visitation $\nu_\pi$ instead of the initial distribution $\zeta$.

By sequentially expanding PVG, we obtain the analytic representation of the policy gradient.

**Analytic Policy Gradient.** From the Bellman equation $V^{\pi_\theta}(s) = \mathbb{E}_\varsigma[(1 - \gamma)r(s, \pi_\theta(s, \varsigma)) + \gamma \mathbb{E}_{\xi^*}[V^{\pi_\theta}(f(s, \pi_\theta(s, \varsigma), \xi^*))]]$, we obtain the following backward recursions:

$$\nabla_\theta V^{\pi_\theta}(s) = \mathbb{E}_\varsigma \Big[ (1 - \gamma)\nabla_a r \nabla_\theta \pi_\theta + \gamma \mathbb{E}_{\xi^*} \big[ \nabla_{s'} V^{\pi_\theta}(s') \nabla_a f \nabla_\theta \pi_\theta + \nabla_\theta V^{\pi_\theta}(s') \big] \Big], \tag{3.2}$$

$$\nabla_s V^{\pi_\theta}(s) = \mathbb{E}_\varsigma \Big[ (1 - \gamma)(\nabla_s r + \nabla_a r \nabla_s \pi_\theta) + \gamma \mathbb{E}_{\xi^*} \big[ \nabla_{s'} V^{\pi_\theta}(s')(\nabla_s f + \nabla_a f \nabla_s \pi_\theta) \big] \Big]. \tag{3.3}$$

See §A for detailed derivations of (3.2) and (3.3). Now we have the RP gradient backpropagated through the transition path starting at $s$. By taking an expectation over the initial state distribution, we obtain the Analytic Policy Gradient (APG) $\nabla_\theta J(\pi_\theta) = \mathbb{E}_{s \sim \zeta}[\nabla_\theta V^{\pi_\theta}(s)]$.

There remain challenges when developing practical algorithms: (1) the above formulas require the gradient information of the transition function $f$. In this work, however, we consider a common RL setting where $f$ is unknown and needs to be fitted by a model. It is thus natural to ask how the properties of the model (e.g., prediction accuracy and model smoothness) affect the gradient estimation and the convergence of the resulting algorithms, and (2) even if we have access to an accurate model, unrolling it over full sequences faces practical difficulties. The memory and computational cost scale linearly with the unroll length. Long chains of nonlinear mappings can also lead to exploding or vanishing gradients and even worse, chaotic phenomenons [9] and difficulty in optimization [43, 36, 58, 38]. These difficulties demand some form of truncation when performing RP PGMs.

---

[1]Due to the simulation property of continuous distributions in (2.3), we interchangeably write $a \sim \pi_\theta(\cdot \mid s)$ with $a = \pi_\theta(s, \varsigma)$ and $s' \sim f(\cdot \mid s, a)$ with $s' = f(s, a, \xi^*)$, where $\xi^*$ is sampled from an unknown distribution.

# 4 Model-Based RP Policy Gradient Methods

Through the application of Model Value Expansion (MVE) for model truncation, this section unveils two RP policy gradient frameworks constructed upon MVE.

## 4.1 $h$-Step Model Value Expansion

To handle the difficulties inherent in full unrolls, many algorithms employ direct truncation, where the long sequence is broken down into short sub-sequences and backpropagation is applied accordingly, e.g., Truncated BPTT [63]. However, such an approach over-prioritizes short-term dependencies, which leads to biased gradient estimates.

In model-based RL (MBRL), one viable solution is to adopt the $h$-step Model Value Expansion [16], which decomposes the value estimation $\widehat{V}^\pi(s)$ into the rewards gleaned from the learned model and a residual estimated by a critic function $\widehat{Q}_\omega$, that is,

$$\widehat{V}^{\pi_\theta}(s) = (1 - \gamma) \cdot \left( \sum_{i=0}^{h-1} \gamma^i \cdot r(\widehat{s}_i, \widehat{a}_i) + \gamma^h \cdot \widehat{Q}_\omega(\widehat{s}_h, \widehat{a}_h) \right),$$

where $\widehat{s}_0 = s$, $\widehat{a}_i = \pi_\theta(\widehat{s}_i, \varsigma)$, and $\widehat{s}_{i+1} = \widehat{f}_\psi(\widehat{s}_i, \widehat{a}_i, \xi)$. Here, the noise variables $\varsigma$ and $\xi$ can be sampled from the fixed distributions or inferred from the real samples, which we now discuss.

## 4.2 Model-Based RP Gradient Estimation

Utilizing the pathwise gradient with respect to $\theta$, we present the following two frameworks.

**Model Derivatives on Predictions (DP).** A straightforward way to compute the first-order gradient is to link the reward, model, policy, and critic together and backpropagate through them. Specifically, the differentiation is carried out on the trajectories simulated by the model $\widehat{f}_\psi$, which serves as a tool for *both* the prediction of states and the evaluation of derivatives. The corresponding RP-DP estimator of gradient $\nabla_\theta J(\pi_\theta)$ is denoted as $\widehat{\nabla}_\theta^{\mathrm{DP}} J(\pi_\theta)$, which takes the form of

$$\widehat{\nabla}_\theta^{\mathrm{DP}} J(\pi_\theta) = \frac{1}{N} \sum_{n=1}^{N} \nabla_\theta \left( \sum_{i=0}^{h-1} \gamma^i \cdot r(\widehat{s}_{i,n}, \widehat{a}_{i,n}) + \gamma^h \cdot \widehat{Q}_\omega(\widehat{s}_{h,n}, \widehat{a}_{h,n}) \right), \tag{4.1}$$

where $\widehat{s}_{0,n} \sim \mu_{\pi_\theta}$, $\widehat{a}_{i,n} = \pi_\theta(\widehat{s}_{i,n}, \varsigma_n)$, and $\widehat{s}_{i+1,n} = \widehat{f}_\psi(\widehat{s}_{i,n}, \widehat{a}_{i,n}, \xi_n)$ with noises $\varsigma_n \sim p(\varsigma)$ and $\xi_n \sim p(\xi)$. Here, $\mu_{\pi_\theta}$ is the distribution where the initial states of the simulated trajectories are sampled. In Section 5, we study a general form of $\mu_{\pi_\theta}$ that is a mixture of the initial state distribution $\zeta$ and the state visitation $\nu_{\pi_\theta}$.

Various algorithms can be instantiated from (4.1) with different choices of $h$. When $h = 0$, the framework reduces to model-free policy gradients, such as RP(0) [4] and the variants of DDPG [34], e.g., SAC [23]. When $h \to \infty$, the resulting algorithm is BPTT [22, 13, 6] where only the model is learned. Recent model-based approaches, such as MAAC [12] and related algorithms [42, 4, 33], require a carefully selected $h$.

**Model Derivatives on Real Samples (DR).** An alternative approach is to use the learned differentiable model solely for the calculation of derivatives, with the aid of Monte-Carlo estimates obtained from *real* samples. By replacing $\nabla_a f, \nabla_s f$ in (3.2)-(3.3) with $\nabla_a \widehat{f}_\psi, \nabla_a \widehat{f}_\psi$ and setting the termination of backpropagation at the $h$-th step as $\widehat{\nabla} V^{\pi_\theta}(\widehat{s}_{h,n}) = \nabla \widehat{V}_\omega(\widehat{s}_{h,n})$, we are able to derive a dynamic representation of $\widehat{\nabla}_\theta V^{\pi_\theta}$, which we defer to §A. The corresponding RP-DR gradient estimator is

$$\widehat{\nabla}_\theta^{\mathrm{DR}} J(\pi_\theta) = \frac{1}{N} \sum_{n=1}^{N} \widehat{\nabla}_\theta V^{\pi_\theta}(\widehat{s}_{0,n}), \tag{4.2}$$

where $\widehat{s}_{0,n} \sim \mu_{\pi_\theta}$. Equation (4.2) can be specified as (A.9), which is in the same format as (4.1), but with the noise variables $\varsigma_n, \xi_n$ inferred from the real data sample $(s_i, a_i, s_{i+1})$ via the relation $a_i = \pi_\theta(s_i, \varsigma_n)$ and $s_{i+1} = \widehat{f}_\psi(s_i, a_i, \xi_n)$ (see §A for details). Algorithms such as SVG [25] and its variants [1, 5] are examples of this RP-DR method.

## 4.3 Algorithmic Framework

The pseudocode of model-based RP PGMs is presented in Algorithm 1, where three update procedures are performed iteratively. In other words, the policy, model, and critic are updated at each iteration $t \in [T]$, generating sequences of $\{\pi_{\theta_t}\}_{t \in [T+1]}$, $\{\widehat{f}_{\psi_t}\}_{t \in [T]}$, and $\{\widehat{Q}_{\omega_t}\}_{t \in [T]}$, respectively.

---

**Algorithm 1** Model-Based Reparameterization Policy Gradient

---

**Input:** Number of iterations $T$, learning rate $\eta$, batch size $N$, empty dataset $\mathcal{D}$
1: **for** iteration $t \in [T]$ **do**
2:  Update the model parameter $\psi_t$ by MSE or MLE
3:  Update the critic parameter $\omega_t$ by performing Temporal Difference
4:  Sample states from $\mu_{\pi_t}$ and estimate $\widehat{\nabla}_\theta J(\pi_{\theta_t}) = \widehat{\nabla}_\theta^{\mathrm{DP}} J(\pi_{\theta_t})$ (4.1) or $\widehat{\nabla}_\theta^{\mathrm{DR}} J(\pi_{\theta_t})$ (4.2)
5:  Update the policy parameter $\theta_t$ by $\theta_{t+1} \leftarrow \theta_t + \eta \cdot \widehat{\nabla}_\theta J(\pi_{\theta_t})$
6:  Execute $\pi_{\theta_{t+1}}$ and save data to $\mathcal{D}$ to obtain $\mathcal{D}_{t+1}$
7: **end for**

---

**Policy Update.** The update rule for the policy parameter $\theta \in \Theta$ with learning rate $\eta$ is as follows,

$$\theta_{t+1} \leftarrow \theta_t + \eta \cdot \widehat{\nabla}_\theta J(\pi_{\theta_t}), \tag{4.3}$$

where $\widehat{\nabla}_\theta J(\pi_{\theta_t})$ can be specified as $\widehat{\nabla}_\theta^{\mathrm{DP}} J(\pi_{\theta_t})$ or $\widehat{\nabla}_\theta^{\mathrm{DR}} J(\pi_{\theta_t})$.

**Model Update.** By predicting the mean of transition with minimized mean squared error (MSE) or fitting a probabilistic model with maximum likelihood estimation (MLE), e.g., $\psi_t = \mathrm{argmax}_{\psi \in \Psi} \mathbb{E}_{\mathcal{D}_t}[\log \widehat{f}_\psi(s_{i+1}|s_i, a_i)]$, canonical MBRL methods learn forward models that predict how the system evolves when an action is taken at a state.

However, accurate state predictions do not imply accurate RP gradient estimation. Thus, we define $\epsilon_f(t)$ to denote the model (gradient) error at iteration $t$:

$$\epsilon_f(t) = \max_{i \in [h]} \mathbb{E}_{\mathbb{P}(s_i, a_i), \mathbb{P}(\widehat{s}_i, \widehat{a}_i)} \left[ \left\| \frac{\partial s_i}{\partial s_{i-1}} - \frac{\partial \widehat{s}_i}{\partial \widehat{s}_{i-1}} \right\|_2 + \left\| \frac{\partial s_i}{\partial a_{i-1}} - \frac{\partial \widehat{s}_i}{\partial \widehat{a}_{i-1}} \right\|_2 \right], \tag{4.4}$$

where $\mathbb{P}(s_i, a_i)$ is the true state-action distribution at the $i$-th timestep by following $s_0 \sim \nu_{\pi_{\theta_t}}$, $a_j \sim \pi_{\theta_t}(\cdot \,|\, s_j)$, $s_{j+1} \sim f(\cdot \,|\, s_j, a_j)$, with policy and transition noise sampled from a fixed distribution. Similarly, $\mathbb{P}(\widehat{s}_i, \widehat{a}_i)$ is the model rollout distribution at the $i$-th timestep by following $\widehat{s}_0 \sim \nu_{\pi_{\theta_t}}$, $\widehat{a}_j \sim \pi_{\theta_t}(\cdot \,|\, \widehat{s}_j)$, $\widehat{s}_{j+1} \sim \widehat{f}_{\psi_t}(\cdot \,|\, \widehat{s}_j, \widehat{a}_j)$, where the noise is sampled when we use RP-DP gradient estimator and is inferred from real samples when we use RP-DR gradient estimator (in this case $\mathbb{P}(\widehat{s}_i, \widehat{a}_i) = \mathbb{P}(s_i, a_i)$).

In MBRL, it is common to learn a state-predictive model that can make multi-step predictions. However, this presents a challenge in reconciling the discrepancy between minimizing state prediction error and the gradient error of the model. Although it is natural to consider regularizing the models' directional derivatives to be consistent with the samples [33], we contend that the use of state-predictive models does *not* cripple our analysis of gradient bias based on $\epsilon_f$: For learned models that extrapolate beyond the visited regions, the gradient error can still be bounded via finite difference. In other words, $\epsilon_f$ can be expressed as the mean squared training error with an additional measure of the model class complexity to capture its generalizability. This same argument can also be applied to the case of learning a critic through temporal difference.

**Critic Update.** For any policy $\pi$, its value function $Q^\pi$ satisfies the Bellman equation, which has a unique solution. In other words, $Q = \mathcal{T}^\pi Q$ if and only if $Q = Q^\pi$. The Bellman operator $\mathcal{T}^\pi$ is defined for any $(s, a) \in \mathcal{S} \times \mathcal{A}$ as

$$\mathcal{T}^\pi Q(s, a) = \mathbb{E}_{\pi, f}\big[(1 - \gamma) \cdot r(s, a) + \gamma \cdot Q(s', a')\big].$$

We aim to approximate the state-action value function $Q^\pi$ with a critic $\widehat{Q}_\omega$. Due to the solution uniqueness of the Bellman equation, it can be achieved by minimizing the mean-squared Bellman error $\omega_t = \mathrm{argmin}_{\omega \in \Omega} \mathbb{E}_{\mathcal{D}_t}[(\widehat{Q}_\omega(s, a) - \mathcal{T}^\pi \widehat{Q}_\omega(s, a))^2]$ via Temporal Difference (TD) [55, 10]. We define the critic error at the $t$-th iteration as follows,

$$\epsilon_v(t) = \alpha^2 \cdot \mathbb{E}_{\mathbb{P}(s_h, a_h), \mathbb{P}(\widehat{s}_h, \widehat{a}_h)} \left[ \left\| \frac{\partial Q^{\pi_{\theta_t}}}{\partial s} - \frac{\partial \widehat{Q}_{\omega_t}}{\partial \widehat{s}} \right\|_2 + \left\| \frac{\partial Q^{\pi_{\theta_t}}}{\partial a} - \frac{\partial \widehat{Q}_{\omega_t}}{\partial \widehat{a}} \right\|_2 \right], \tag{4.5}$$

where $\alpha = (1 - \gamma)/\gamma^h$ and $\mathbb{P}(s_h, a_h)$, $\mathbb{P}(\widehat{s}_h, \widehat{a}_h)$ are distributions at timestep $h$ with the same definition as in (4.4). The inclusion of $\alpha^2$ ensures that the critic error remains in alignment with the single-step model error $\epsilon_f$: (1) the critic estimates the tail terms that occur after $h$ steps in the model expansion, therefore the step-average critic error should be inversely proportional to the tail discount summation $\sum_{i=h}^{\infty} \gamma^i = 1/\alpha$, and (2) the quadratic form shares similarities with the canonical MBRL analysis – the cumulative error of the model trajectories scales linearly with the single-step prediction error and quadratically with the considered horizon (i.e., tail after the $h$-th step). This is because the cumulative error is linear in the considered horizon and the maximum state discrepancy, which is linear in the single-step error and, again, the horizon [28].

## 5 Main Results

In what follows, we present our main theoretical results, whose detailed proofs are deferred to §B. Specifically, we analyze the convergence of model-based RP PGMs and, more importantly, study the correlation between the convergence rate, gradient bias, variance, smoothness of the model, and approximation error. Based on our theory, we propose various algorithmic designs for MB RP PGMs.

To begin with, we impose a common regularity condition on the policy functions following previous works [68, 46, 69, 3]. The assumption below essentially ensures the smoothness of the objective $J(\pi_\theta)$, which is required by most existing analyses of policy gradient methods [60, 6, 2].

**Assumption 5.1** (Lipschitz and Bounded Score Function). We assume that the score function of policy $\pi_\theta$ is Lipschitz continuous and has bounded norm $\forall (s, a) \in \mathcal{S} \times \mathcal{A}$, that is,

$$\big\| \log \pi_{\theta_1}(a \,|\, s) - \log \pi_{\theta_2}(a \,|\, s) \big\|_2 \le L_1 \cdot \|\theta_1 - \theta_2\|_2, \quad \big\| \log \pi_\theta(a \,|\, s) \big\|_2 \le B_\theta.$$

We characterize the convergence of RP PGMs by first providing the following proposition.

**Proposition 5.2** (Convergence to Stationary Point). We define the gradient bias $b_t$ and variance $v_t$ as

$$b_t = \big\| \nabla_\theta J(\pi_{\theta_t}) - \mathbb{E}\big[ \widehat{\nabla}_\theta J(\pi_{\theta_t}) \big] \big\|_2, \quad v_t = \mathbb{E}\big[ \big\| \widehat{\nabla}_\theta J(\pi_{\theta_t}) - \mathbb{E}\big[ \widehat{\nabla}_\theta J(\pi_{\theta_t}) \big] \big\|_2^2 \big].$$

Suppose the absolute value of the reward $r(s, a)$ is bounded by $|r(s, a)| \le r_{\mathrm{m}}$ for $(s, a) \in \mathcal{S} \times \mathcal{A}$. Let $\delta = \sup \|\theta\|_2$, $L = r_{\mathrm{m}} \cdot L_1/(1 - \gamma)^2 + (1 + \gamma) \cdot r_{\mathrm{m}} \cdot B_\theta^2/(1 - \gamma)^3$, and $c = (\eta - L\eta^2)^{-1}$. It then holds for $T \ge 4L^2$ that

$$\min_{t \in [T]} \mathbb{E}\big[ \big\| \nabla_\theta J(\pi_{\theta_t}) \big\|_2^2 \big] \le \frac{4c}{T} \cdot \mathbb{E}\big[ J(\pi_{\theta_T}) - J(\pi_{\theta_1}) \big] + \frac{4}{T} \bigg( \sum_{t=0}^{T-1} c\big(2\delta \cdot b_t + \frac{\eta}{2} \cdot v_t\big) + b_t^2 + v_t \bigg).$$

Proposition 5.2 illustrates the interdependence between the convergence and the variance, bias of the gradient estimators. In order for model-based RP PGMs to converge, it is imperative to maintain both the variance and bias at sublinear growth rates. Prior to examining the upper bound of $b_t$ and $v_t$, we make the following Lipschitz assumption, which has been implemented in a plethora of preceding studies [46, 12, 33].

**Assumption 5.3** (Lipschitz Continuity). We assume that $r(s, a)$ and $f(s, a, \xi^*)$ are $L_r$ and $L_f$ Lipschitz continuous, respectively. Formally, for any $s_1, s_2 \in \mathcal{S}$, $a_1, a_2 \in \mathcal{A}$, and $\xi^*$,

$$\big| r(s_1, a_1) - r(s_2, a_2) \big| \le L_r \cdot \big\| (s_1 - s_2, a_1 - a_2) \big\|_2,$$
$$\big\| f(s_1, a_1, \xi^*) - f(s_2, a_2, \xi^*) \big\|_2 \le L_f \cdot \big\| (s_1 - s_2, a_1 - a_2) \big\|_2.$$

Let $\widetilde{L}_g = \max\{L_g, 1\}$, where $L_g$ is the Lipschitz of function $g$. We have the following result for gradient variance.

**Proposition 5.4** (Gradient Variance). Under Assumption 5.3, for any $t \in [T]$, the gradient variance of the estimator $\widehat{\nabla}_\theta J(\pi_\theta)$, which can be specified as $\widehat{\nabla}_\theta^{\mathrm{DP}} J(\pi_\theta)$ or $\widehat{\nabla}_\theta^{\mathrm{DR}} J(\pi_\theta)$, can be bounded by

$$v_t = O\bigg( h^4 \bigg( \frac{1 - \gamma^h}{1 - \gamma} \bigg)^2 \widetilde{L}_{\widehat{f}}^{4h} \widetilde{L}_\pi^{4h}/N + \gamma^{2h} h^4 \widetilde{L}_{\widehat{f}}^{4h} \widetilde{L}_\pi^{4h}/N \bigg),$$

where $L_{\widehat{f}} = \sup_{\psi \in \Psi, s_1, s_2 \in \mathcal{S}, a_1, a_2 \in \mathcal{A}} \|\widehat{f}_\psi(s_1, a_1, \xi) - \widehat{f}_\psi(s_2, a_2, \xi)\|_2/\|(s_1 - s_2, a_1 - a_2)\|_2$ and $L_\pi = \sup_{\theta \in \Theta, s_1, s_2 \in \mathcal{S}, \varsigma} \|\pi_\theta(s_1, \varsigma) - \pi_\theta(s_2, \varsigma)\|_2/\|s_1 - s_2\|_2$.

We observe that the variance upper bound exhibits a polynomial dependence on the Lipschitz continuity of the model and policy, where the degrees are linear in the model unroll length. This makes sense intuitively, as the transition can be highly chaotic when $L_{\widehat{f}} > 1$ and $L_\pi > 1$. This can result in diverging trajectories and variable gradient directions during training, leading to significant variance in the gradients.

**Remark 5.5.** Model-based RP PGMs with non-smooth models and policies can suffer from large variance and highly non-smooth loss landscapes, which can lead to slow convergence or failure during training even in simple toy examples [42, 37, 53]. Proposition 5.4 suggests that one can add smoothness regularization to avoid exploding gradient variance. See our discussion at the end of this section for more details.

*Model-based* RP PGMs possess unique advantages by utilizing proxy models for variance reduction. By enforcing the smoothness of the model, the gradient variance is reduced without a burden when the underlying transition is smooth. However, in cases of non-smooth dynamics, doing so may introduce additional bias due to increased model estimation error. This necessitates a trade-off between the model error and gradient variance. Nevertheless, our empirical study demonstrates that smoothness regularization improves performance in robotic locomotion tasks, despite the cost of increased bias.

Next, we study the gradient bias. We consider the case where the state distribution $\mu_\pi$, which is used for estimating the RP gradient, is a mixture of the initial distribution $\zeta$ of the MDP and the state visitation $\nu_\pi$. In other words, we consider $\mu_\pi = \beta \cdot \nu_\pi + (1 - \beta) \cdot \zeta$, where $\beta \in [0, 1]$. This form is of particular interest as it encompasses various state sampling schemes that can be employed, such as when $h = 0$ and $h \to \infty$: When not utilizing a model, such as in SVG(0) [25, 4] and DDPG [34], states are sampled from $\nu_\pi$; while when unrolling the model over full sequences, as in BPTT, states are sampled from the initial distribution.

Given that the effects of policy actions extend to all future states and rewards, unless we know the exact policy value function, its gradient $\nabla_\theta Q^{\pi_\theta}$ cannot be simply represented by quantities in any finite timescale. Hence, the differentiation of the critic function does not align with the true value gradient that has recursive structures. To tackle this issue, we provide the gradient bias bound that is based on the measure of discrepancy between the initial distribution $\zeta$ and the state visitation $\nu_\pi$.

**Proposition 5.6** (Gradient Bias). We denote $\kappa = \sup_\pi \mathbb{E}_{\nu_\pi}[(\mathrm{d}\zeta/\mathrm{d}\nu_\pi(s))^2]^{1/2}$, where $\mathrm{d}\zeta/\mathrm{d}\nu_\pi$ is the Radon-Nikodym derivative of $\zeta$ with respect to $\nu_\pi$. Let $\kappa' = \beta + \kappa \cdot (1 - \beta)$. Under Assumption 5.3, for any $t \in [T]$, the gradient bias is bounded by

$$b_t = O\Big(\kappa\kappa' h^2 (1 - \gamma^h)\widetilde{L}_{\widehat{f}}^h \widetilde{L}_f^h \widetilde{L}_\pi^{2h} \epsilon_{f,t}/(1 - \gamma) + \kappa' h \gamma^{3h} \widetilde{L}_{\widehat{f}}^h \widetilde{L}_\pi^h \epsilon_{v,t}/(1 - \gamma)^2\Big),$$

where $\epsilon_{f,t}$ and $\epsilon_{v,t}$ are the shorthand notations of $\epsilon_f(t)$ defined in (4.4) and $\epsilon_v(t)$ in (4.5), respectively.

The analysis above yields the identification of an optimal model expansion step $h^*$ that achieves the best convergence rate, whose form is presented by the following proposition.

**Proposition 5.7** (Optimal Model Expansion Step). Given $L_f \leq 1$, if we regularize the model and policy so that $L_{\widehat{f}} \leq 1$ and $L_\pi \leq 1$, then when $\gamma \approx 1$, the optimal model expansion step $h^*$ at iteration $t$ that minimizes the convergence rate upper bound satisfies $h^* = \max\{h'^*, 0\}$, where $h'^* = O(\epsilon_{v,t}/((1 - \gamma)(\epsilon_{f,t} + \epsilon_{v,t})))$ scales linearly with $\epsilon_{v,t}/(\epsilon_{f,t} + \epsilon_{v,t})$ and the effective task horizon $1/(1 - \gamma)$.

In Proposition 5.7, the Lipschitz condition of the underlying dynamics, i.e., $L_f \leq 1$, ensures the stability of the system. This can be seen in the linear system example, where the transitions are determined by the eigenspectrum of the family of transformations, leading to exponential divergence of trajectories w.r.t. the largest eigenvalue. In cases where this condition is not met in practical control systems, finding the best model unroll length may require trial and error. Fortunately, we have observed through experimentation that enforcing smoothness offers a much wider range of unrolling lengths that still provide satisfactory results.

**Remark 5.8.** As the error scale $\epsilon_{v,t}/(\epsilon_{f,t} + \epsilon_{v,t})$ increases, so too does the value of $h^*$. This finding can inform the practical algorithms to rely more on the model by performing longer unrolls when the model error $\epsilon_{f,t}$ is small, while avoiding long unrolls when the critic error $\epsilon_{v,t}$ is small.

**A Spectral Normalization Method.** To ensure a smooth transition and faster convergence, we propose using a Spectral Normalization (SN) [39] model-based RP PGM that applies SN to all layers of the deep model network and policy network. While other techniques, such as adversarial regularization [50], exist, we focus primarily on SN as it directly regulates the Lipschitz constant of the function. Specifically, the Lipschitz constant $L_g$ of a function $g$ satisfies $L_g = \sup_x \sigma_{\max}(\nabla g(x))$, where $\sigma_{\max}(W)$ denotes the largest singular value of the matrix $W$, defined as $\sigma_{\max}(W) = \max_{\|x\|_2 \le 1} \|Wx\|_2$. For neural network $f$ with linear layers $g(x) = W_i x$ and 1-Lipschitz activation (e.g., ReLU and leaky ReLU), we have $L_g = \sigma_{\max}(W_i)$ and $L_{\widehat{f}} \le \prod_i \sigma_{\max}(W_i)$. By normalizing the spectral norm of $W_i$ with $W_i^{\text{SN}} = W_i / \sigma_{\max}(W_i)$, SN guarantees that the Lipschitz of $f$ is upper-bounded by 1.

Finally, we characterize the algorithm convergence rate.

**Corollary 5.9** (Convergence Rate). Let $\varepsilon(T) = \sum_{t=0}^{T-1} b_t$. We have for $T \ge 4L^2$ that
$$\min_{t \in [T]} \mathbb{E}\left[\left\|\nabla_\theta J(\pi_{\theta_t})\right\|_2^2\right] \le 16\delta \cdot \varepsilon(T)/\sqrt{T} + 4\varepsilon^2(T)/T + O\left(1/\sqrt{T}\right).$$

The convergence rate can be further clarified by determining how quickly the errors of model and critic approach zero, i.e., $\sum_{t=0}^{T-1} \epsilon_f(t) + \epsilon_v(t)$. Such results can be accomplished by conducting a more fine-grained investigation of the model and critic function classes, such as utilizing overparameterized neural nets with width scaling with $T$ to bound the training error, as done in [10, 35], and incorporating complexity measures of the model and critic function classes to bound $\epsilon_f(t)$ and $\epsilon_v(t)$. Their forms, however, are beyond the scope of this paper.

# 6 Related Work

**Policy Gradient Methods.** Within the RL field, the LR estimator is the basis of most policy gradient algorithms, e.g., REINFORCE [64] and actor-critic methods [56, 31, 30, 14]. Recent works [3, 60, 7, 35] have shown the global convergence of LR policy gradient under certain conditions, while less attention has been focused on RP PGMs. Remarkably, the analysis in [33] is based on the strong assumptions on the *chained* gradient and ignores the impact of value approximation, which oversimplifies the problem by reducing the $h$-step model value expansion to single-step model unrolls. Besides, [12] *only* focused on the gradient bias while still neglecting the necessary visitation analysis. Despite the utilization in our method of Spectral Normalization on the learned model to control the gradient variance, SN has also been applied in deep RL to *value* functions in order to enable deeper neural nets [8] or regulate the value-aware model error [72].

**Differentiable Simulation.** This paper delves into the model-based setting [11, 28, 29, 70, 24], where a learned model is employed to train a control policy. Recent approaches [41, 53, 54, 67] based on differentiable simulators [18, 27] assume that gradients of simulation outcomes w.r.t. actions are explicitly given. To deal with the discontinuities and empirical bias phenomenon in the differentiable simulation caused by contact dynamics, previous works proposed smoothing the gradient adaptively with a contact-aware central-path parameter [71], using penalty-based contact formulations [20, 66] or adopting randomized smoothing for hard-contact dynamics [53, 54]. However, these are not in direct comparison to our analysis, which relies on model function approximators.

# 7 Experiments

## 7.1 Evaluation of Reparameterization Policy Gradient Methods

To gain a deeper understanding and support the theoretical findings, we evaluate several algorithms originating from our RP PGM framework and compare them with various baselines in Figure 1. Specifically, RP-DP-SN is the proposed SN-based algorithm; RP-DP, as described in (4.1), is implemented as MAAC [12] with entropy regularization [4]; RP-DR, as described in (4.2), is implemented as SVG [25]; the model-free RP(0) is described in §4.2. Details and discussions are deferred to §C.

The results indicate that RP-DP consistently outperforms or matches the performance of existing methods such as MBPO [28] and LR PGMs, including REINFORCE [56], NPG [31], ACKTR [65], and PPO [49]. This highlights the significance and potential of model-based RP PGMs. Due to space limitations, we refer the readers to §C.5 for larger versions of the figures in the experiment section.

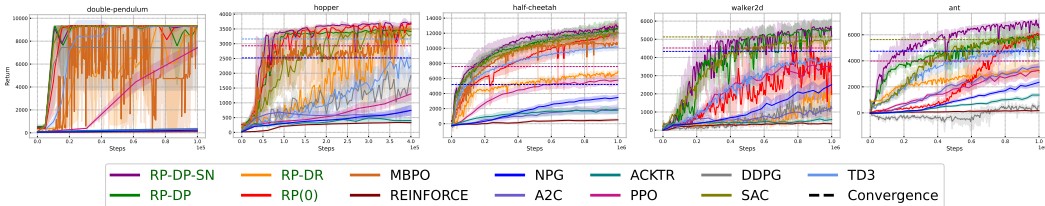

Figure 1: Comparisons between RP PGMs (the green labels) and MF/MB baselines (the black labels) in the MuJoCo [57] tasks.

## 7.2 Gradient Variance and Loss Landscape

Our prior investigations have revealed that vanilla MB RP PGMs tend to have highly non-smooth landscapes due to the significant increase in gradient variance. We now conduct experiments to validate this phenomenon in practice. In Figure 2, we plot the mean gradient variance of the vanilla RP-DP algorithm during training. To visualize the loss landscapes, we plot in Figure 3 the negative value estimate along two directions that are randomly selected in the policy parameter space of a training policy.

We can observe that for vanilla RP policy gradient algorithms, the gradient variance explodes in exponential rate with respect to the model unroll length. This results in a loss landscape that is highly non-smooth for larger unrolling steps. This renders the importance of smoothness regularization. Specifically, incorporating Spectral Normalization (SN) [39] in

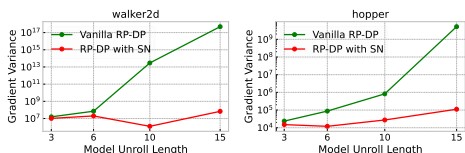

Figure 2: Gradient variance of the vanilla RP-DP explodes while adding spectral normalization solves this issue.

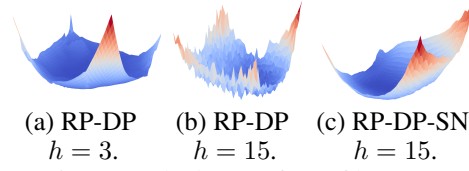

(a) RP-DP $h = 3$.    (b) RP-DP $h = 15$.    (c) RP-DP-SN $h = 15$.

Figure 3: The loss surface of hopper.

the model and policy neural nets leads to a marked reduction in mean gradient variance for all unroll length settings, resulting in a much smoother loss surface compared to the vanilla implementation.

## 7.3 Benefit of Smoothness Regularization

In this section, we investigate the effect of smoothness regularization to support our claim: The gradient variance has polynomial dependence on the Lipschitz continuity of the model and policy, which is a contributing factor to training. Our results in Figure 4 show that SN-based RP PGMs achieve equivalent or superior performance compared to the vanilla implementation. Importantly, for longer model unrolls (e.g., 10 in walker2d and 15 in hopper), vanilla RP PGMs fail to produce reliable performance. SN-based methods, on the other hand, significantly boost training.

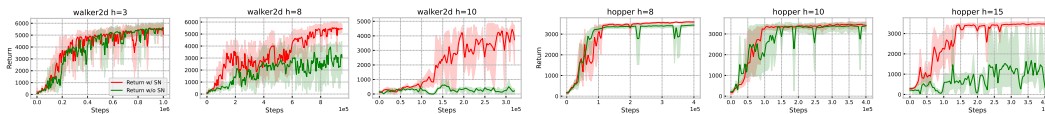

Figure 4: Performance of vanilla and SN-based MB RP PGMs with varying $h$. The vanilla method only works with a small $h$ and fails when $h$ increases, while the SN-based method enables a larger $h$.

Additionally, we explore different choices of model unroll lengths and examine the impact of spectral normalization, with results shown in Figure 5. We find that by utilizing SN, the curse of chaos can be mitigated, allowing for longer model unrolls. This is crucial for practical algorithmic designs: The most popular model-based RP PGMs such as [12, 4] often rely on a carefully chosen (small) $h$ (e.g., $h = 3$). When the model is good enough, a small $h$ may not

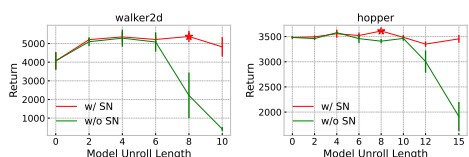

Figure 5: Performance with different $h$.

fully leverage the accurate gradient information. As evidence, approaches [67, 41] based on differentiable simulators typically adopt longer unrolls compared to model-based approaches. Therefore, with SN, more accurate multi-step predictions should enable more efficient learning without making the

underlying optimization process harder. SN-based approaches also provide more robustness since the return is insensitive to $h$ and the variance of return is smaller compared to the vanilla implementation when $h$ is large.

**Ablation on Variance.** By plotting the gradient variance of RP-DP during training in Figure 6, we can discern that for walker $h = 10$ and hopper $h = 15$, a key contributor to the failure of vanilla RP-DP is the exploding gradient variances. On the contrary, the SN-based approach excels in training performance as a result of the drastically reduced variance.

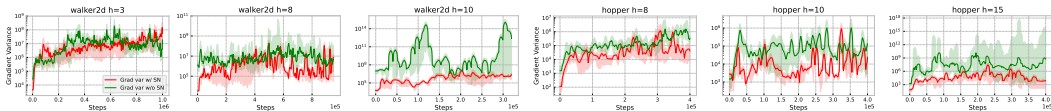

Figure 6: Gradient variance of RP PGMs. The variance is significantly lower with SN when $h$ is large.

**Ablation on Bias.** When the underlying MDP is itself contact-rich and has non-smooth or even discontinuous dynamics, explicitly regularizing the Lipschitz of the transition model may lead to large error $\epsilon_f$ and thus large gradient bias. Therefore, it is also important to study if SN causes such a negative effect and if it does, how to trade off between the model bias and gradient variance. To efficiently obtain an accurate first-order gradient (instead of via finite difference in MuJoCo), we conduct ablation based on the *differentiable* simulator dFlex [26, 67], where Analytic Policy Gradient (APG) described in Section 3 can be implemented.

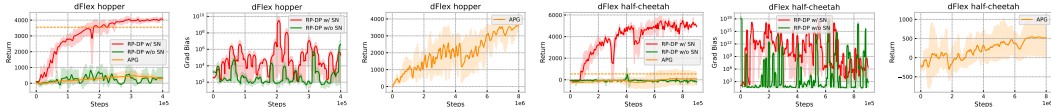

Figure 7: Performance and gradient bias in differentiable simulation. The third and last columns are the full training curves of APG, which need 20 times more steps than RP-DP-SN to reach a comparable return in the hopper task and fail in the half-cheetah task, respectively.

Figure 7 illustrates the crucial role SN plays in locomotion tasks. It is worth noting that the higher bias of the SN method does *not* impede performance, but rather improves it, indicating that the primary obstacle in training RP PGMs is the large variance in gradients. Therefore, even if the simulation is differentiable, learning a smooth proxy model can be beneficial when the dynamics have bumps or discontinuous jumps, which is usually the case in robotics systems, sharing similarities with the gradient smoothing techniques [53, 54, 71] for APG.

# 8 Conclusion & Future Work

In this work, we study the convergence of model-based reparameterization policy gradient methods and identify the determining factors that affect the quality of gradient estimation. Based on our theory, we propose a spectral normalization (SN) method to mitigate the exploding gradient variance issue. Our experimental results also support the proposed theory and method. Since SN-based RP PGMs allow longer model unrolls without introducing additional optimization hardness, learning more accurate multi-step models to fully leverage their gradient information should be a fruitful future direction. It will also be interesting to explore different smoothness regularization designs and apply them to a broader range of algorithms, such as using proxy models in differentiable simulation to obtain smooth policy gradients, which we would like to leave as future work.

# Acknowledgements

The authors would like to thank Yan Li for his valuable insights and discussions during the early stages of this paper. Zhaoran Wang acknowledges National Science Foundation (Awards 2048075, 2008827, 2015568, 1934931), Simons Institute (Theory of Reinforcement Learning), Amazon, J.P. Morgan, and Two Sigma for their supports.

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

# A  Recursive Expression of Analytic Policy Gradient

In what follows, we interchangeably write $\nabla_a x$ and $\mathrm{d}x/\mathrm{d}a$ as the gradient and denote by $\partial x/\partial a$ the partial derivative.

**Derivation of Analytic Policy Gradient.** First of all, we provide the derivation of (3.2) and (3.3), i.e., the backward recursions of the gradient in APG.

Following [25], we define the operator

$$\nabla_\theta^i = \sum_{j \geq i} \frac{\mathrm{d}a_j}{\mathrm{d}\theta} \cdot \frac{\partial}{\partial a_j} + \sum_{j > i} \frac{\mathrm{d}s_j}{\mathrm{d}\theta} \cdot \frac{\partial}{\partial s_j}. \tag{A.1}$$

We begin by expanding the total derivative operator by chain rule as

$$\begin{aligned}
\frac{\mathrm{d}}{\mathrm{d}\theta} &= \sum_{i \geq 0} \frac{\mathrm{d}a_i}{\mathrm{d}\theta} \cdot \frac{\partial}{\partial a_i} + \sum_{i > 0} \frac{\mathrm{d}s_i}{\mathrm{d}\theta} \cdot \frac{\partial}{\partial s_i} \\
&= \frac{\mathrm{d}a_0}{\mathrm{d}\theta} \cdot \frac{\partial}{\partial a_0} + \frac{\mathrm{d}s_1}{\mathrm{d}\theta} \cdot \frac{\partial}{\partial s_1} + \sum_{i \geq 1} \frac{\mathrm{d}a_i}{\mathrm{d}\theta} \cdot \frac{\partial}{\partial a_i} + \sum_{i > 1} \frac{\mathrm{d}s_i}{\mathrm{d}\theta} \cdot \frac{\partial}{\partial s_i}.
\end{aligned} \tag{A.2}$$

Here, the expansion holds when $\mathrm{d}/\mathrm{d}\theta$ operates on policies and models that are differentiable with respect to all states $s_i$ and actions $a_i$.

Plugging (A.2) into (A.1), we obtain the following recursive formula for $\nabla_\theta^i$,

$$\nabla_\theta^i = \frac{\mathrm{d}a_i}{\mathrm{d}\theta} \cdot \frac{\partial}{\partial a_i} + \frac{\mathrm{d}a_t}{\mathrm{d}\theta} \cdot \frac{\mathrm{d}s_{i+1}}{\mathrm{d}a_t} \cdot \frac{\partial}{\partial s_{i+1}} + \nabla_\theta^{i+1}. \tag{A.3}$$

By the Bellman equation, we have

$$V^{\pi_\theta}(s) = \mathbb{E}_\varsigma \left[ (1-\gamma) \cdot r(s, \pi_\theta(s, \varsigma)) + \gamma \cdot \mathbb{E}_{\xi^*} \left[ V^{\pi_\theta}\left( f\left(s, \pi_\theta(s, \varsigma), \xi^*\right)\right)\right]\right]. \tag{A.4}$$

Combining (A.3) and (A.4) gives

$$\begin{aligned}
\nabla_\theta V^{\pi_\theta}(s) = \frac{\mathrm{d}V^{\pi_\theta}(s)}{\mathrm{d}\theta} &= \frac{\mathrm{d}}{\mathrm{d}\theta} \mathbb{E}_\varsigma \left[ (1-\gamma) \cdot r(s, \pi_\theta(s, \varsigma)) + \gamma \cdot \mathbb{E}_{\xi^*} \left[ V^{\pi_\theta}\left( f\left(s, \pi_\theta(s, \varsigma), \xi^*\right)\right)\right]\right] \\
&= \mathbb{E}_\varsigma \left[ (1-\gamma) \cdot \frac{\partial r}{\partial a} \cdot \frac{\mathrm{d}a}{\mathrm{d}\theta} + \gamma \cdot \mathbb{E}_{\xi^*} \left[ \frac{\mathrm{d}a}{\mathrm{d}\theta} \cdot \frac{\partial s'}{\partial a} \cdot \frac{\mathrm{d}V^{\pi_\theta}(s')}{\mathrm{d}s'} + \frac{\mathrm{d}V^{\pi_\theta}(s')}{\mathrm{d}\theta}\right]\right],
\end{aligned} \tag{A.5}$$

which corresponds to (3.2).

For the $\mathrm{d}V^\pi(s)/\mathrm{d}s$ term on the right-hand side of (A.5), we have the following recursion,

$$\begin{aligned}
\nabla_s V^{\pi_\theta}(s) = \frac{\mathrm{d}V^{\pi_\theta}(s)}{\mathrm{d}s} &= \frac{\mathrm{d}}{\mathrm{d}s} \mathbb{E}_\varsigma \left[ (1-\gamma) \cdot r(s, \pi_\theta(s, \varsigma)) + \gamma \cdot \mathbb{E}_{\xi^*} \left[ V^{\pi_\theta}\left( f\left(s, \pi_\theta(s, \varsigma), \xi^*\right)\right)\right]\right] \\
&= \mathbb{E}_\varsigma \left[ (1-\gamma) \cdot \left( \frac{\partial r}{\partial s} + \frac{\partial r}{\partial a} \cdot \frac{\partial a}{\partial s}\right) + \gamma \cdot \mathbb{E}_{\xi^*} \left[ \frac{\partial s'}{\partial s} \cdot \frac{\mathrm{d}V^{\pi_\theta}(s')}{\mathrm{d}s'} + \frac{\partial s'}{\partial a} \cdot \frac{\partial a}{\partial s} \cdot \frac{\mathrm{d}V^{\pi_\theta}(s')}{\mathrm{d}s'}\right]\right],
\end{aligned} \tag{A.6}$$

which corresponds to (3.3).

Therefore, we complete the derivative of (3.2) and (3.3).

**Derivation of RP-DR Policy Gradient.** By the same arguments in (A.5) and (A.6) with $\nabla_a f$ (or $\partial s'/\partial a$) and $\nabla_s f$ (or $\partial s'/\partial s$) replaced with $\nabla_a \widehat{f}_\psi$ and $\nabla_s \widehat{f}_\psi$, we obtain

$$\begin{aligned}
\widehat{\nabla}_\theta V^{\pi_\theta}(\widehat{s}_{i,n}) &= (1-\gamma)\nabla_a r(\widehat{s}_{i,n}, \widehat{a}_{i,n})\nabla_\theta \pi_\theta(\widehat{s}_{i,n}, \varsigma_n) \\
&\quad + \gamma \widehat{\nabla}_s V^{\pi_\theta}(\widehat{s}_{i+1,n})\nabla_a \widehat{f}_\psi(\widehat{s}_{i,n}, \widehat{a}_{i,n}, \xi_n)\nabla_\theta \pi_\theta(\widehat{s}_{i,n}, \varsigma_n) + \gamma \widehat{\nabla}_\theta V^{\pi_\theta}(\widehat{s}_{i+1,n}),
\end{aligned} \tag{A.7}$$

$$\begin{aligned}
\widehat{\nabla}_s V^{\pi_\theta}(\widehat{s}_{i,n}) &= (1-\gamma)\left(\nabla_s r(\widehat{s}_{i,n}, \widehat{a}_{i,n}) + \nabla_a r(\widehat{s}_{i,n}, \widehat{a}_{i,n})\nabla_s \pi_\theta(\widehat{s}_{i,n}, \varsigma_n)\right) \\
&\quad + \gamma \widehat{\nabla}_s V^{\pi_\theta}(\widehat{s}_{i+1,n})\left(\nabla_s \widehat{f}_\psi(\widehat{s}_{i,n}, \widehat{a}_{i,n}, \xi_n) + \nabla_a \widehat{f}_\psi(\widehat{s}_{i,n}, \widehat{a}_{i,n}, \xi_n)\nabla_s \pi_\theta(\widehat{s}_{i,n}, \varsigma_n)\right),
\end{aligned} \tag{A.8}$$

where the termination of backpropagation at the $h$-th step is $\widehat{\nabla} V^{\pi_\theta}(\widehat{s}_{h,n}) = \nabla \widehat{V}_\omega(\widehat{s}_{h,n})$.

Combining (A.7) and (A.8), we obtain the RP-DR policy gradient estimator as follows,

$$\widehat{\nabla}_\theta^{\mathrm{DR}} J(\pi_\theta) = \frac{1}{N} \sum_{n=1}^{N} \widehat{\nabla}_\theta V^{\pi_\theta}(\widehat{s}_{0,n}) = \frac{1}{N} \sum_{n=1}^{N} \nabla_\theta \left( \sum_{i=0}^{h-1} \gamma^i r(\widehat{s}_{i,n}, \widehat{a}_{i,n}) + \gamma^h \widehat{Q}_\omega(\widehat{s}_{h,n}, \widehat{a}_{h,n}) \right),$$
(A.9)

where $\widehat{s}_{0,n} \sim \mu_{\pi_\theta}$, $\widehat{a}_{i,n} = \pi_\theta(\widehat{s}_{i,n}, \varsigma_n)$, and $\widehat{s}_{i+1,n} = \widehat{f}_\psi(\widehat{s}_{i,n}, \widehat{a}_{i,n}, \xi_n)$. Here, $\varsigma_n$ and $\xi_n$ are inferred by solving $a_i = \pi_\theta(s_i, \varsigma_n)$ and $s_{i+1} = \widehat{f}_\psi(s_i, a_i, \xi_n)$, respectively, where $(s_i, a_i, s_{i+1})$ is the real data sample. For example, for a state $s_{i+1}$ sampled from a one-dimensional Gaussian transition model $s_{i+1} \sim \mathcal{N}(\phi(s_i, a_i), \sigma^2)$, where the variance is $\sigma$ and the mean $\phi(s_i, a_i)$ is the output of some function parameterized by $\phi$, the noise $\xi_n$ can be inferred as $\xi_n = (s_{i+1} - \phi(s_i, a_i))/\sigma$.

# B    Proofs

## B.1    Proof of Proposition 5.2

As a preparation before proving Proposition 5.2, we first present the following lemma stating that the objective in (2.1) is Lipschitz smooth under Assumption 5.1.

**Lemma B.1** (Smooth Objective, [69] Lemma 3.2). The objective $J(\pi_\theta)$ is $L$-smooth in $\theta$, such that $\|\nabla_\theta J(\pi_{\theta_1}) - \nabla_\theta J(\pi_{\theta_2})\|_2 \le L\|\theta_1 - \theta_2\|_2$, where

$$L = \frac{r_{\mathrm{m}} \cdot L_1}{(1-\gamma)^2} + \frac{(1+\gamma) \cdot r_{\mathrm{m}} \cdot B_\theta^2}{(1-\gamma)^3}.$$

Then we are ready to prove Proposition 5.2.

*Proof of Proposition 5.2.* See the proof of Theorem 4.2 in [71]. □

## B.2    Proof of Proposition 5.4

*Proof.* Since the RP-DP gradient $\widehat{\nabla}_\theta J(\pi_\theta) = \widehat{\nabla}_\theta^{\mathrm{DP}} J(\pi_\theta)$ in (4.1) and the RP-DR gradient $\widehat{\nabla}_\theta J(\pi_\theta) = \widehat{\nabla}_\theta^{\mathrm{DR}} J(\pi_\theta)$ in (4.2) share the same state transition $\widehat{s}_{i+1,n} = \widehat{f}(\widehat{s}_{i,n}, \xi_n)$, where recall that the only difference lies in the source of noise $\xi_n$, our subsequent analysis holds for both RP-DP and RP-DR.

To upper-bound the gradient variance $v_t = \mathbb{E}[\|\widehat{\nabla}_\theta J(\pi_{\theta_t}) - \mathbb{E}[\widehat{\nabla}_\theta J(\pi_{\theta_t})]\|_2^2]$, we characterize the norm inside the outer expectation.

We start with the case where the sample size $N = 1$, which naturally generalizes to $N > 1$. Specifically, we consider an *arbitrary* $h$-step trajectory obtained by unrolling the model under policy $\pi_{\theta_t}$. We denote the pathwise gradient $\widehat{\nabla}_\theta J(\pi_{\theta_t})$ of this trajectory as $g'$. Then we have

$$v_t \le \max_{g'} \left\| g' - \mathbb{E}[\widehat{\nabla}_\theta J(\pi_{\theta_t})] \right\|_2^2 = \left\| g - \mathbb{E}[\widehat{\nabla}_\theta J(\pi_{\theta_t})] \right\|_2^2 = \left\| \mathbb{E}[g - \widehat{\nabla}_\theta J(\pi_{\theta_t})] \right\|_2^2,$$

where $g$ is the pathwise gradient $\widehat{\nabla}_\theta J(\pi_{\theta_t})$ of a *fixed* (but unknown) trajectory $(\widehat{s}_{0,n}, \widehat{a}_{0,n}, \widehat{s}_{1,n}, \widehat{a}_{1,n}, \cdots)$ such that the maximum is achieved.

Using the fact that $\|\mathbb{E}[\cdot]\|_2 \le \mathbb{E}[\|\cdot\|_2]$, we further obtain

$$v_t \le \mathbb{E}\left[ \left\| g - \widehat{\nabla}_\theta J(\pi_{\theta_t}) \right\|_2 \right]^2.$$
(B.1)

Let $\widehat{x}_{i,n} = (\widehat{s}_{i,n}, \widehat{a}_{i,n})$. By the triangular inequality, we have

$$\mathbb{E}\left[ \left\| g - \widehat{\nabla}_\theta J(\pi_{\theta_t}) \right\|_2 \right] \le \sum_{i=0}^{h-1} \gamma^i \cdot \mathbb{E}_{\overline{x}_i}\left[ \left\| \nabla_\theta r(\widehat{x}_{i,n}) - \nabla_\theta r(\overline{x}_i) \right\|_2 \right]$$

$$+ \gamma^h \cdot \mathbb{E}_{\overline{x}_h}\left[ \left\| \nabla \widehat{Q}(\widehat{x}_{h,n}) \nabla_\theta \widehat{x}_{h,n} - \nabla \widehat{Q}(\overline{x}_h) \nabla_\theta \overline{x}_h \right\|_2 \right].$$
(B.2)

By the chain rule, we have for any $i \geq 1$ that

$$\frac{\mathrm{d}\widehat{a}_{i,n}}{\mathrm{d}\theta} = \frac{\partial\widehat{a}_{i,n}}{\partial\widehat{s}_{i,n}} \cdot \frac{\mathrm{d}\widehat{s}_{i,n}}{\mathrm{d}\theta} + \frac{\partial\widehat{a}_{i,n}}{\partial\theta}, \tag{B.3}$$

$$\frac{\mathrm{d}\widehat{s}_{i,n}}{\mathrm{d}\theta} = \frac{\partial\widehat{s}_{i,n}}{\partial\widehat{s}_{i-1,n}} \cdot \frac{\mathrm{d}\widehat{s}_{i-1,n}}{\mathrm{d}\theta} + \frac{\partial\widehat{s}_{i,n}}{\partial\widehat{a}_{i-1,n}} \cdot \frac{\mathrm{d}\widehat{a}_{i-1,n}}{\mathrm{d}\theta}. \tag{B.4}$$

Denote by $L_\theta = \sup_{\theta\in\Theta, s\in\mathcal{S}, \varsigma} \|\nabla_\theta \pi_\theta(s, \varsigma)\|_2$.

Plugging $\mathrm{d}\widehat{a}_{i-1,n}/\mathrm{d}\theta$ in (B.3) into (B.4), we get

$$\left\|\frac{\mathrm{d}\widehat{s}_{i,n}}{\mathrm{d}\theta}\right\|_2 = \left\|\left(\frac{\partial\widehat{s}_{i,n}}{\partial\widehat{s}_{i-1,n}} + \frac{\partial\widehat{s}_{i,n}}{\partial\widehat{a}_{i-1,n}} \cdot \frac{\partial\widehat{a}_{i-1,n}}{\partial\widehat{s}_{i-1,n}}\right) \cdot \frac{\mathrm{d}\widehat{s}_{i-1,n}}{\mathrm{d}\theta} + \frac{\partial\widehat{s}_{i,n}}{\partial\widehat{a}_{i-1,n}} \cdot \frac{\partial\widehat{a}_{i-1,n}}{\partial\theta}\right\|_2$$
$$\leq L_{\widehat{f}}\widetilde{L}_\pi \cdot \left\|\frac{\mathrm{d}\widehat{s}_{i-1,n}}{\mathrm{d}\theta}\right\|_2 + L_{\widehat{f}}L_\theta, \tag{B.5}$$

where the inequality follows from Assumption 5.3 and the Cauchy-Schwarz inequality.

Recursively applying (B.5), we obtain for any $i \geq 1$ that

$$\left\|\frac{\mathrm{d}\widehat{s}_{i,n}}{\mathrm{d}\theta}\right\|_2 \leq L_{\widehat{f}}L_\theta \cdot \sum_{j=0}^{i-1} L_{\widehat{f}}^j \widetilde{L}_\pi^j \leq i \cdot L_\theta L_{\widehat{f}}^{i+1}\widetilde{L}_\pi^i, \tag{B.6}$$

where the first inequality follows from the induction

$$z_i = az_{i-1} + b = a \cdot (az_{i-2} + b) + b = a^i \cdot z_0 + b \cdot \sum_{j=0}^{i-1} a^j. \tag{B.7}$$

In (B.7), $\{z_j\}_{0\leq j\leq i}$ is the real sequence satisfying $z_j = az_{j-1} + b$. For $\mathrm{d}\widehat{a}_{i,n}/\mathrm{d}\theta$ defined in (B.3), we further have

$$\left\|\frac{\mathrm{d}\widehat{a}_{i,n}}{\mathrm{d}\theta}\right\|_2 \leq L_\pi \cdot \left\|\frac{\mathrm{d}\widehat{s}_{i,n}}{\mathrm{d}\theta}\right\|_2 + L_\theta \leq i \cdot L_\theta L_{\widehat{f}}^{i+1}\widetilde{L}_\pi^{i+1} + L_\theta. \tag{B.8}$$

Combining (B.6) and (B.8), we obtain

$$\left\|\frac{\mathrm{d}\widehat{x}_{i,n}}{\mathrm{d}\theta}\right\|_2 = \left\|\frac{\mathrm{d}\widehat{s}_{i,n}}{\mathrm{d}\theta}\right\|_2 + \left\|\frac{\mathrm{d}\widehat{a}_{i,n}}{\mathrm{d}\theta}\right\|_2 \leq \underbrace{2i \cdot L_\theta L_{\widehat{f}}^{i+1}\widetilde{L}_\pi^{i+1} + L_\theta}_{\widehat{K}(i)}. \tag{B.9}$$

Therefore, we bound the second term on the right-hand side of (B.2) as follows,

$$\mathbb{E}_{\overline{x}_h}\left[\left\|\nabla\widehat{Q}(\widehat{x}_{h,n})\nabla_\theta\widehat{x}_{h,n} - \nabla\widehat{Q}(\overline{x}_h)\nabla_\theta\overline{x}_h\right\|_2\right]$$
$$\leq \mathbb{E}_{\overline{x}_h}\left[\left\|\nabla\widehat{Q}(\widehat{x}_{h,n})\nabla_\theta\widehat{x}_{h,n} - \nabla\widehat{Q}(\overline{x}_h)\nabla_\theta\widehat{x}_{h,n}\right\|_2\right] + \mathbb{E}_{\overline{x}_h}\left[\left\|\nabla\widehat{Q}(\overline{x}_h)\nabla_\theta\widehat{x}_{h,n} - \nabla\widehat{Q}(\overline{x}_h)\nabla_\theta\overline{x}_h\right\|_2\right]$$
$$\leq 2L_{\widehat{Q}} \cdot \widehat{K}(i) + L_{\widehat{Q}} \cdot \left(\mathbb{E}_{\overline{s}_i}\left[\left\|\frac{\mathrm{d}\widehat{s}_{i,n}}{\mathrm{d}\theta} - \frac{\mathrm{d}\overline{s}_i}{\mathrm{d}\theta}\right\|_2\right] + \mathbb{E}_{\overline{a}_i}\left[\left\|\frac{\mathrm{d}\widehat{a}_{i,n}}{\mathrm{d}\theta} - \frac{\mathrm{d}\overline{a}_i}{\mathrm{d}\theta}\right\|_2\right]\right), \tag{B.10}$$

where the last inequality follows from the Cauchy-Schwartz inequality and Assumption 5.3, and $L_{\widehat{Q}} = \sup_{\omega\in\Omega, s_1, s_2\in\mathcal{S}, a_1, a_2\in\mathcal{A}} |\widehat{Q}_\omega(s_1, a_1) - \widehat{Q}_\omega(s_2, a_2)|/\|(s_1 - s_2, a_1 - a_2)\|_2$.

By the chain rule, we bound the first term on the right-hand side of (B.2) as follows,

$$\mathbb{E}_{\overline{x}_i}\left[\left\|\nabla_\theta r(\widehat{x}_{i,n}) - \nabla_\theta r(\overline{x}_i)\right\|_2\right]$$
$$= \mathbb{E}_{\overline{x}_i}\left[\left\|\nabla r(\widehat{x}_{i,n})\nabla_\theta\widehat{x}_{i,n} - \nabla r(\overline{x}_i)\nabla_\theta\overline{x}_i\right\|_2\right]$$
$$\leq \mathbb{E}_{\overline{x}_i}\left[\left\|\nabla r(\widehat{x}_{i,n})\nabla_\theta\widehat{x}_{i,n} - \nabla r(\widehat{x}_{i,n})\nabla_\theta\overline{x}_i\right\|_2\right] + \mathbb{E}\left[\left\|\nabla r(\widehat{x}_{i,n})\nabla_\theta\overline{x}_i - \nabla r(\overline{x}_i)\nabla_\theta\overline{x}_i\right\|_2\right]$$
$$\leq L_r \cdot \left(\mathbb{E}_{\overline{s}_i}\left[\left\|\frac{\mathrm{d}\widehat{s}_{i,n}}{\mathrm{d}\theta} - \frac{\mathrm{d}\overline{s}_i}{\mathrm{d}\theta}\right\|_2\right] + \mathbb{E}_{\overline{a}_i}\left[\left\|\frac{\mathrm{d}\widehat{a}_{i,n}}{\mathrm{d}\theta} - \frac{\mathrm{d}\overline{a}_i}{\mathrm{d}\theta}\right\|_2\right]\right) + 2L_r \cdot \widehat{K}(i). \tag{B.11}$$

Plugging (B.10) and (B.11) into (B.2) and (B.1), we obtain

$$v_t \leq \left[ \left( L_r \cdot \sum_{i=0}^{h-1} \gamma^i + \gamma^h \cdot L_{\widehat{Q}} \right) \cdot \left( \mathbb{E}_{\overline{s}_h} \left[ \left\| \frac{\mathrm{d}\widehat{s}_{h,n}}{\mathrm{d}\theta} - \frac{\mathrm{d}\overline{s}_h}{\mathrm{d}\theta} \right\|_2 \right] + \mathbb{E}_{\overline{a}_h} \left[ \left\| \frac{\mathrm{d}\widehat{a}_{h,n}}{\mathrm{d}\theta} - \frac{\mathrm{d}\overline{a}_h}{\mathrm{d}\theta} \right\|_2 \right] + 2\widehat{K}(h) \right) \right]^2$$

$$= O\left( h^4 \left( \frac{1-\gamma^h}{1-\gamma} \right)^2 \widetilde{L}_{\widehat{f}}^{4h} \widetilde{L}_{\pi}^{4h} + \gamma^{2h} h^4 \widetilde{L}_{\widehat{f}}^{4h} \widetilde{L}_{\pi}^{4h} \right), \tag{B.12}$$

where the inequality follows from Lemma B.2 and by plugging the definition of $\widehat{K}(i)$ in (B.9).

Note that the variance $v_t$ scales with the batch size $N$ at the rate of $1/N$. Since the analysis above is established for $N = 1$, the bound of the gradient variance $v_t$ is established by dividing the right-hand side of (B.12) by $N$, which concludes the proof of Proposition 5.4. $\qquad \square$

**Lemma B.2.** Denote $e = \sup \mathbb{E}_{\overline{s}_0}[\|\mathrm{d}\widehat{s}_{0,n}/\mathrm{d}\theta - \mathrm{d}\overline{s}_0/\mathrm{d}\theta\|_2]$, which is a constant that only depends on the initial state distribution[2]. For any timestep $i \geq 1$ and the corresponding state, action, we have the following results,

$$\mathbb{E}_{\overline{s}_i} \left[ \left\| \frac{\mathrm{d}\widehat{s}_{i,n}}{\mathrm{d}\theta} - \frac{\mathrm{d}\overline{s}_i}{\mathrm{d}\theta} \right\|_2 \right] \leq \widetilde{L}_{\widehat{f}}^i \widetilde{L}_{\pi}^i \left( e + 4i \cdot \widetilde{L}_{\widehat{f}} \widetilde{L}_{\pi} \cdot \widehat{K}(i-1) + 2i \cdot \widetilde{L}_{\widehat{f}} L_\theta \right),$$

$$\mathbb{E}_{\overline{a}_i} \left[ \left\| \frac{\mathrm{d}\widehat{a}_{i,n}}{\mathrm{d}\theta} - \frac{\mathrm{d}\overline{a}_i}{\mathrm{d}\theta} \right\|_2 \right] \leq \widetilde{L}_{\widehat{f}}^i \widetilde{L}_{\pi}^{i+1} \left( e + 4i \cdot \widetilde{L}_{\widehat{f}} \widetilde{L}_{\pi} \cdot \widehat{K}(i-1) + 2i \cdot \widetilde{L}_{\widehat{f}} L_\theta \right) + 2L_\pi \widehat{K}(i) + 2L_\theta.$$

*Proof.* Firstly, from (B.4), we obtain for any $i \geq 1$ that

$$\mathbb{E}_{\overline{s}_i} \left[ \left\| \frac{\mathrm{d}\widehat{s}_{i,n}}{\mathrm{d}\theta} - \frac{\mathrm{d}\overline{s}_i}{\mathrm{d}\theta} \right\|_2 \right]$$

$$= \mathbb{E} \left[ \left\| \frac{\partial \widehat{s}_{i,n}}{\partial \widehat{s}_{i-1,n}} \cdot \frac{\mathrm{d}\widehat{s}_{i-1,n}}{\mathrm{d}\theta} + \frac{\partial \widehat{s}_{i,n}}{\partial \widehat{a}_{i-1,n}} \cdot \frac{\mathrm{d}\widehat{a}_{i-1,n}}{\mathrm{d}\theta} - \frac{\partial \overline{s}_i}{\partial \overline{s}_{i-1}} \cdot \frac{\mathrm{d}\overline{s}_{i-1}}{\mathrm{d}\theta} - \frac{\partial \overline{s}_i}{\partial \overline{a}_{i-1}} \cdot \frac{\mathrm{d}\overline{a}_{i-1}}{\mathrm{d}\theta} \right\|_2 \right]$$

According to the triangle inequality, we further have

$$\leq \mathbb{E} \left[ \left\| \frac{\partial \widehat{s}_{i,n}}{\partial \widehat{s}_{i-1,n}} \cdot \frac{\mathrm{d}\widehat{s}_{i-1,n}}{\mathrm{d}\theta} - \frac{\partial \overline{s}_i}{\partial \overline{s}_{i-1}} \cdot \frac{\mathrm{d}\widehat{s}_{i-1,n}}{\mathrm{d}\theta} \right\|_2 \right] + \mathbb{E} \left[ \left\| \frac{\partial \overline{s}_i}{\partial \overline{s}_{i-1}} \cdot \frac{\mathrm{d}\widehat{s}_{i-1,n}}{\mathrm{d}\theta} - \frac{\partial \overline{s}_i}{\partial \overline{s}_{i-1}} \cdot \frac{\mathrm{d}\overline{s}_{i-1}}{\mathrm{d}\theta} \right\|_2 \right]$$

$$+ \mathbb{E} \left[ \left\| \frac{\partial \widehat{s}_{i,n}}{\partial \widehat{a}_{i-1,n}} \cdot \frac{\mathrm{d}\widehat{a}_{i-1,n}}{\mathrm{d}\theta} - \frac{\partial \overline{s}_i}{\partial \overline{a}_{i-1}} \cdot \frac{\mathrm{d}\widehat{a}_{i-1,n}}{\mathrm{d}\theta} \right\|_2 \right] + \mathbb{E} \left[ \left\| \frac{\partial \overline{s}_i}{\partial \overline{a}_{i-1}} \cdot \frac{\mathrm{d}\widehat{a}_{i-1,n}}{\mathrm{d}\theta} - \frac{\partial \overline{s}_i}{\partial \overline{a}_{i-1}} \cdot \frac{\mathrm{d}\overline{a}_{i-1}}{\mathrm{d}\theta} \right\|_2 \right]$$

$$\leq 2L_{\widehat{f}} \cdot \left( \left\| \frac{\mathrm{d}\widehat{s}_{i-1,n}}{\mathrm{d}\theta} \right\|_2 + \left\| \frac{\mathrm{d}\widehat{a}_{i-1,n}}{\mathrm{d}\theta} \right\|_2 \right) + L_{\widehat{f}} \cdot \mathbb{E}_{\overline{s}_{i-1}} \left[ \left\| \frac{\mathrm{d}\widehat{s}_{i-1,n}}{\mathrm{d}\theta} - \frac{\mathrm{d}\overline{s}_{i-1}}{\mathrm{d}\theta} \right\|_2 \right]$$

$$+ L_{\widehat{f}} \cdot \mathbb{E}_{\overline{a}_{i-1}} \left[ \left\| \frac{\mathrm{d}\widehat{a}_{i-1,n}}{\mathrm{d}\theta} - \frac{\mathrm{d}\overline{a}_{i-1}}{\mathrm{d}\theta} \right\|_2 \right]. \tag{B.13}$$

Similarly, we have from (B.3) that

$$\mathbb{E}_{\overline{a}_i} \left[ \left\| \frac{\mathrm{d}\widehat{a}_{i,n}}{\mathrm{d}\theta} - \frac{\mathrm{d}\overline{a}_i}{\mathrm{d}\theta} \right\|_2 \right]$$

$$= \mathbb{E} \left[ \left\| \frac{\partial \widehat{a}_{i,n}}{\partial \widehat{s}_{i,n}} \cdot \frac{\mathrm{d}\widehat{s}_{i,n}}{\mathrm{d}\theta} + \frac{\partial \widehat{a}_{i,n}}{\partial \theta} - \frac{\partial \overline{a}_i}{\partial \overline{s}_i} \cdot \frac{\mathrm{d}\overline{s}_i}{\mathrm{d}\theta} - \frac{\partial \overline{a}_i}{\partial \theta} \right\|_2 \right]$$

$$\leq \mathbb{E} \left[ \left\| \frac{\partial \widehat{a}_{i,n}}{\partial \widehat{s}_{i,n}} \cdot \frac{\mathrm{d}\widehat{s}_{i,n}}{\mathrm{d}\theta} - \frac{\partial \overline{a}_i}{\partial \overline{s}_i} \cdot \frac{\mathrm{d}\widehat{s}_{i,n}}{\mathrm{d}\theta} \right\|_2 \right] + \mathbb{E} \left[ \left\| \frac{\partial \overline{a}_i}{\partial \overline{s}_i} \cdot \frac{\mathrm{d}\widehat{s}_{i,n}}{\mathrm{d}\theta} - \frac{\partial \overline{a}_i}{\partial \overline{s}_i} \cdot \frac{\mathrm{d}\overline{s}_i}{\mathrm{d}\theta} \right\|_2 \right] + \mathbb{E} \left[ \left\| \frac{\partial \widehat{a}_{i,n}}{\partial \theta} - \frac{\partial \overline{a}_i}{\partial \theta} \right\|_2 \right]$$

$$\leq 2L_\pi \cdot \mathbb{E} \left[ \left\| \frac{\mathrm{d}\widehat{s}_{i,n}}{\mathrm{d}\theta} \right\| \right] + L_\pi \cdot \mathbb{E} \left[ \left\| \frac{\mathrm{d}\widehat{s}_{i,n}}{\mathrm{d}\theta} - \frac{\mathrm{d}\overline{s}_i}{\mathrm{d}\theta} \right\|_2 \right] + 2L_\theta. \tag{B.14}$$

---

[2]We define $e$ to account for the stochasticity of the initial state distribution. $e = 0$ when the initial state is deterministic.

Plugging (B.14) back to (B.13), we obtain

$$\mathbb{E}_{\overline{s}_i}\left[\left\|\frac{\mathrm{d}\widehat{s}_{i,n}}{\mathrm{d}\theta} - \frac{\mathrm{d}\overline{s}_i}{\mathrm{d}\theta}\right\|_2\right]$$

$$\lesssim 4L_{\widehat{f}}\widetilde{L}_\pi \cdot \left(\left\|\frac{\mathrm{d}\widehat{s}_{i-1,n}}{\mathrm{d}\theta}\right\|_2 + \left\|\frac{\mathrm{d}\widehat{a}_{i-1,n}}{\mathrm{d}\theta}\right\|_2\right) + L_{\widehat{f}}\widetilde{L}_\pi \cdot \mathbb{E}_{\overline{s}_{i-1}}\left[\left\|\frac{\mathrm{d}\widehat{s}_{i-1,n}}{\mathrm{d}\theta} - \frac{\mathrm{d}\overline{s}_{i-1}}{\mathrm{d}\theta}\right\|_2\right] + 2L_{\widehat{f}}L_\theta$$

$$\leq 4L_{\widehat{f}}\widetilde{L}_\pi \cdot \widehat{K}(i-1) + L_{\widehat{f}}\widetilde{L}_\pi \cdot \mathbb{E}_{\overline{s}_{i-1}}\left[\left\|\frac{\mathrm{d}\widehat{s}_{i-1,n}}{\mathrm{d}\theta} - \frac{\mathrm{d}\overline{s}_{i-1}}{\mathrm{d}\theta}\right\|_2\right] + 2L_{\widehat{f}}L_\theta,$$

where the last inequality follows from the definition of $\widehat{K}$ in (B.9).

Applying this recursion gives

$$\mathbb{E}_{\overline{s}_i}\left[\left\|\frac{\mathrm{d}\widehat{s}_{i,n}}{\mathrm{d}\theta} - \frac{\mathrm{d}\overline{s}_i}{\mathrm{d}\theta}\right\|_2\right] \leq e\big(L_{\widehat{f}}\widetilde{L}_\pi\big)^i + \big(4L_{\widehat{f}}\widetilde{L}_\pi \cdot \widehat{K}(i-1) + 2L_{\widehat{f}}L_\theta\big) \cdot \sum_{j=0}^{i-1}\big(L_{\widehat{f}}\widetilde{L}_\pi\big)^j$$

$$\leq \widetilde{L}_{\widehat{f}}^i \widetilde{L}_\pi^i\Big(e + 4i \cdot \widetilde{L}_{\widehat{f}}\widetilde{L}_\pi \cdot \widehat{K}(i-1) + 2i \cdot \widetilde{L}_{\widehat{f}}L_\theta\Big),$$

where the first equality follows from (B.7).

As a consequence, we have from (B.14) that

$$\mathbb{E}_{\overline{a}_i}\left[\left\|\frac{\mathrm{d}\widehat{a}_{i,n}}{\mathrm{d}\theta} - \frac{\mathrm{d}\overline{a}_i}{\mathrm{d}\theta}\right\|_2\right] \leq \widetilde{L}_{\widehat{f}}^i \widetilde{L}_\pi^{i+1}\Big(e + 4i \cdot \widetilde{L}_{\widehat{f}}\widetilde{L}_\pi \cdot \widehat{K}(i-1) + 2i \cdot \widetilde{L}_{\widehat{f}}L_\theta\Big) + 2L_\pi\widehat{K}(i) + 2L_\theta.$$

This concludes the proof. $\square$

## B.3 Proof of Proposition 5.6

*Proof.* The analysis of gradient bias differs from that of gradient variance as it involves not only the distribution of approximate states but also the recurrent dependencies of the true value on future timesteps, which must be given extra attention.

In the following analysis, we will first apply similar techniques as those outlined in the previous section to establish an upper bound on the decomposed reward terms in the gradient bias. Afterward, we will address the distribution mismatch issue caused by the recursive structure of $V^{\pi_\theta}$ and the non-recursive structure of the value approximation $\widehat{V}_{\omega_t}$.

**Step 1: Bound the cumulative reward terms in the gradient bias.**

To begin with, we decompose the bias of the reward gradient at timestep $i \geq 0$ as follows,

$$\mathbb{E}_{(s_i,a_i)\sim\mathbb{P}(s_i,a_i),(\widehat{s}_{i,n},\widehat{a}_{i,n})\sim\mathbb{P}(\widehat{s}_i,\widehat{a}_i)}\left[\left\|\frac{\mathrm{d}r(\widehat{x}_{i,n})}{\mathrm{d}\theta} - \frac{\mathrm{d}r(x_i)}{\mathrm{d}\theta}\right\|_2\right]$$

$$= \mathbb{E}\left[\left\|\frac{\mathrm{d}r}{\mathrm{d}\widehat{x}_{i,n}} \cdot \frac{\mathrm{d}\widehat{x}_{i,n}}{\mathrm{d}\theta} - \frac{\mathrm{d}r}{\mathrm{d}x_i} \cdot \frac{\mathrm{d}x_i}{\mathrm{d}\theta}\right\|_2\right]$$

$$\leq \mathbb{E}\left[\left\|\frac{\mathrm{d}r}{\mathrm{d}\widehat{x}_{i,n}} \cdot \frac{\mathrm{d}\widehat{x}_{i,n}}{\mathrm{d}\theta} - \frac{\mathrm{d}r}{\mathrm{d}x_i} \cdot \frac{\mathrm{d}\widehat{x}_{i,n}}{\mathrm{d}\theta}\right\|_2 + \left\|\frac{\mathrm{d}r}{\mathrm{d}x_i} \cdot \frac{\mathrm{d}\widehat{x}_{i,n}}{\mathrm{d}\theta} - \frac{\mathrm{d}r}{\mathrm{d}x_i} \cdot \frac{\mathrm{d}x_i}{\mathrm{d}\theta}\right\|_2\right]$$

$$\leq 2L_r \cdot \widehat{K}(i) + L_r \cdot \left(\mathbb{E}\left[\left\|\frac{\mathrm{d}\widehat{s}_{i,n}}{\mathrm{d}\theta} - \frac{\mathrm{d}s_i}{\mathrm{d}\theta}\right\|_2\right] + \mathbb{E}\left[\left\|\frac{\mathrm{d}\widehat{a}_{i,n}}{\mathrm{d}\theta} - \frac{\mathrm{d}a_i}{\mathrm{d}\theta}\right\|_2\right]\right), \quad (B.15)$$

where $\mathbb{P}(s_i,a_i)$ and $\mathbb{P}(\widehat{s}_i,\widehat{a}_i)$ are defined in (4.4) with respect to $s_0 \sim \nu_\pi$, $\widehat{s}_0 \sim \nu_\pi$.

We have from (B.3) that for any $i \geq 1$,

$$\mathbb{E}\left[\left\|\frac{\mathrm{d}\widehat{a}_{i,n}}{\mathrm{d}\theta} - \frac{\mathrm{d}a_i}{\mathrm{d}\theta}\right\|_2\right]$$

$$= \mathbb{E}\left[\left\|\frac{\partial\widehat{a}_{i,n}}{\partial\widehat{s}_{i,n}} \cdot \frac{\mathrm{d}\widehat{s}_{i,n}}{\mathrm{d}\theta} + \frac{\partial\widehat{a}_{i,n}}{\partial\theta} - \frac{\partial a_i}{\partial s_i} \cdot \frac{\mathrm{d}s_i}{\mathrm{d}\theta} - \frac{\partial a_i}{\partial\theta}\right\|_2\right]$$

By the triangle inequality and the Lipschitz assumption, it then follows that

$$\leq \mathbb{E}\left[\left\|\frac{\partial \widehat{a}_{i,n}}{\partial \widehat{s}_{i,n}} \cdot \frac{\mathrm{d}\widehat{s}_{i,n}}{\mathrm{d}\theta} - \frac{\partial a_i}{\partial s_i} \cdot \frac{\mathrm{d}\widehat{s}_{i,n}}{\mathrm{d}\theta}\right\|_2\right] + \mathbb{E}\left[\left\|\frac{\partial a_i}{\partial s_i} \cdot \frac{\mathrm{d}\widehat{s}_{i,n}}{\mathrm{d}\theta} - \frac{\partial a_i}{\partial s_i} \cdot \frac{\mathrm{d}s_i}{\mathrm{d}\theta}\right\|_2\right] + \mathbb{E}\left[\left\|\frac{\partial \widehat{a}_{i,n}}{\partial \theta} - \frac{\partial a_i}{\partial \theta}\right\|_2\right]$$

$$\leq 2L_\pi \cdot \mathbb{E}\left[\left\|\frac{\mathrm{d}\widehat{s}_{i,n}}{\mathrm{d}\theta}\right\|_2\right] + L_\pi \cdot \mathbb{E}\left[\left\|\frac{\mathrm{d}\widehat{s}_{i,n}}{\mathrm{d}\theta} - \frac{\mathrm{d}s_i}{\mathrm{d}\theta}\right\|_2\right] + 2L_\theta. \tag{B.16}$$

Similarly, we have from (B.4) that for any $i \geq 1$,

$$\mathbb{E}\left[\left\|\frac{\mathrm{d}\widehat{s}_{i,n}}{\mathrm{d}\theta} - \frac{\mathrm{d}s_i}{\mathrm{d}\theta}\right\|_2\right]$$

$$= \mathbb{E}\left[\left\|\frac{\partial \widehat{s}_{i,n}}{\partial \widehat{s}_{i-1,n}} \cdot \frac{\mathrm{d}\widehat{s}_{i-1,n}}{\mathrm{d}\theta} + \frac{\partial \widehat{s}_{i,n}}{\partial \widehat{a}_{i-1,n}} \cdot \frac{\mathrm{d}\widehat{a}_{i-1,n}}{\mathrm{d}\theta} - \frac{\partial s_i}{\partial s_{i-1}} \cdot \frac{\mathrm{d}s_{i-1}}{\mathrm{d}\theta} - \frac{\partial s_i}{\partial a_{i-1}} \cdot \frac{\mathrm{d}a_{i-1}}{\mathrm{d}\theta}\right\|_2\right]$$

Applying the triangle inequality to extract the $\epsilon_{f,t}$ term defined in (4.4), we proceed by

$$\leq \mathbb{E}\left[\left\|\frac{\partial \widehat{s}_{i,n}}{\partial \widehat{s}_{i-1,n}} \cdot \frac{\mathrm{d}\widehat{s}_{i-1,n}}{\mathrm{d}\theta} - \frac{\partial s_i}{\partial s_{i-1}} \cdot \frac{\mathrm{d}\widehat{s}_{i-1,n}}{\mathrm{d}\theta}\right\|_2\right] + \mathbb{E}\left[\left\|\frac{\partial s_i}{\partial s_{i-1}} \cdot \frac{\mathrm{d}\widehat{s}_{i-1,n}}{\mathrm{d}\theta} - \frac{\partial s_i}{\partial s_{i-1}} \cdot \frac{\mathrm{d}s_{i-1}}{\mathrm{d}\theta}\right\|_2\right]$$

$$+ \mathbb{E}\left[\left\|\frac{\partial \widehat{s}_{i,n}}{\partial \widehat{a}_{i-1,n}} \cdot \frac{\mathrm{d}\widehat{a}_{i-1,n}}{\mathrm{d}\theta} - \frac{\partial s_i}{\partial a_{i-1}} \cdot \frac{\mathrm{d}\widehat{a}_{i-1,n}}{\mathrm{d}\theta}\right\|_2\right] + \mathbb{E}\left[\left\|\frac{\partial s_i}{\partial a_{i-1}} \cdot \frac{\mathrm{d}\widehat{a}_{i-1,n}}{\mathrm{d}\theta} - \frac{\partial s_i}{\partial a_{i-1}} \cdot \frac{\mathrm{d}a_{i-1}}{\mathrm{d}\theta}\right\|_2\right]$$

$$\leq \epsilon_{f,t} \cdot \mathbb{E}\left[\left\|\frac{\mathrm{d}\widehat{s}_{i-1,n}}{\mathrm{d}\theta}\right\|_2 + \left\|\frac{\mathrm{d}\widehat{a}_{i-1,n}}{\mathrm{d}\theta}\right\|_2\right] + L_f \cdot \mathbb{E}\left[\left\|\frac{\mathrm{d}\widehat{s}_{i-1,n}}{\mathrm{d}\theta} - \frac{\mathrm{d}s_{i-1}}{\mathrm{d}\theta}\right\|_2\right]$$

$$+ L_f \cdot \mathbb{E}\left[\left\|\frac{\mathrm{d}\widehat{a}_{i-1,n}}{\mathrm{d}\theta} - \frac{\mathrm{d}a_{i-1}}{\mathrm{d}\theta}\right\|_2\right]$$

$$\leq \epsilon_{f,t} \cdot \widehat{K}(i-1) + L_f \cdot \mathbb{E}\left[\left\|\frac{\mathrm{d}\widehat{s}_{i-1,n}}{\mathrm{d}\theta} - \frac{\mathrm{d}s_{i-1}}{\mathrm{d}\theta}\right\|_2\right] + L_f \cdot \mathbb{E}\left[\left\|\frac{\mathrm{d}\widehat{a}_{i-1,n}}{\mathrm{d}\theta} - \frac{\mathrm{d}a_{i-1}}{\mathrm{d}\theta}\right\|_2\right], \tag{B.17}$$

where the last inequality follows from the definition of $\widehat{K}(i-1)$ in (B.9).

Combining (B.16) and (B.17), we have

$$\mathbb{E}\left[\left\|\frac{\mathrm{d}\widehat{s}_{i,n}}{\mathrm{d}\theta} - \frac{\mathrm{d}s_i}{\mathrm{d}\theta}\right\|_2\right] \lesssim (\epsilon_{f,t} + 2L_f L_\pi) \cdot \widehat{K}(i-1) + L_f \widetilde{L}_\pi \cdot \mathbb{E}\left[\left\|\frac{\mathrm{d}\widehat{s}_{i-1,n}}{\mathrm{d}\theta} - \frac{\mathrm{d}s_{i-1}}{\mathrm{d}\theta}\right\|_2\right] + 2L_f L_\theta$$

$$= \left((\epsilon_{f,t} + 2L_f L_\pi) \cdot \widehat{K}(i-1) + 2L_f L_\theta\right) \cdot \sum_{j=0}^{i-1} L_f^j \widetilde{L}_\pi^j$$

$$\leq \left((\epsilon_{f,t} + 2L_f L_\pi) \cdot \widehat{K}(i-1) + 2L_f L_\theta\right) \cdot i \cdot \widetilde{L}_f^i \widetilde{L}_\pi^i, \tag{B.18}$$

where the last inequality follows from (B.7) and the fact that $s_0, \widehat{s}_{0,n}$ are sampled from the same initial distribution, and the equality holds by applying the recursion.

Plugging (B.18) into (B.16), we obtain

$$\mathbb{E}\left[\left\|\frac{\mathrm{d}\widehat{a}_{i,n}}{\mathrm{d}\theta} - \frac{\mathrm{d}a_i}{\mathrm{d}\theta}\right\|_2\right] \leq \left[(\epsilon_{f,t} + 2L_f L_\pi) \cdot \widehat{K}(i-1) + 2L_f L_\theta\right] \cdot i \cdot \widetilde{L}_f^i \widetilde{L}_\pi^{i+1} + 2L_\pi \widehat{K}(i) + 2L_\theta. \tag{B.19}$$

### Step 2: Address the state distribution mismatch issue.

The next step is to address the distribution mismatch issue caused by the recursive structure of the value function and the non-recursive structure of the value approximation, i.e., the critic.

We define $\overline{\sigma}_1(s,a) = \mathbb{P}(s_h = s, a_h = a)$ where $s_0 \sim \nu_\pi$, $a_i \sim \pi(\cdot \,|\, s_i)$, and $s_{i+1} \sim f(\cdot \,|\, s_i, a_i)$. In a similar way, we define $\widehat{\sigma}_1(s,a) = \mathbb{P}(\widehat{s}_h = s, \widehat{a}_h = a)$ where $\widehat{s}_0 \sim \nu_\pi$, $\widehat{a}_i \sim \pi(\cdot \,|\, \widehat{s}_i)$, and $\widehat{s}_{i+1} \sim \widehat{f}(\cdot \,|\, \widehat{s}_i, \widehat{a}_i)$.

Now we are ready to bound the gradient bias. From Lemma B.4, we know that

$$
b_t \leq \kappa\kappa' \cdot \mathbb{E}_{s_0 \sim \nu_\pi, \widehat{s}_{0,n} \sim \nu_\pi} \left[ \left\| \nabla_\theta \sum_{i=0}^{h-1} \gamma^i \cdot r(s_i, a_i) - \nabla_\theta \sum_{i=0}^{h-1} \gamma^i \cdot r(\widehat{s}_{i,n}, \widehat{a}_{i,n}) \right\|_2 \right]
$$
$$
+ \kappa' \gamma^h \cdot \mathbb{E}_{(s_h, a_h) \sim \overline{\sigma}_1, (\widehat{s}_{h,n}, \widehat{a}_{h,n}) \sim \widehat{\sigma}_1} \left[ \left\| \frac{\partial Q^{\pi_\theta}}{\partial a_h} \cdot \frac{\mathrm{d}a_h}{\mathrm{d}\theta} + \frac{\partial Q^{\pi_\theta}}{\partial s_h} \cdot \frac{\mathrm{d}s_h}{\mathrm{d}\theta} - \nabla_\theta \widehat{Q}_t(\widehat{s}_{h,n}, \widehat{a}_{h,n}) \right\|_2 \right],
$$
(B.20)

where recall that $\kappa' = \beta + \kappa \cdot (1 - \beta)$.

From Lemma B.5, we have for any policy $\pi_\theta$ that the state-action value function is $L_Q$-Lipschitz continuous, which gives for any $s \in \mathcal{S}$ and $a \in \mathcal{A}$ that

$$
\left\| \frac{\partial Q^{\pi_\theta}}{\partial a} \right\|_2 \leq L_Q, \quad \left\| \frac{\partial Q^{\pi_\theta}}{\partial s} \right\|_2 \leq L_Q. \tag{B.21}
$$

The bias brought by the critic, i.e., the last term on the right-hand side of (B.20), can be further bounded by

$$
\mathbb{E}_{(s_h, a_h) \sim \overline{\sigma}_1, (\widehat{s}_{h,n}, \widehat{a}_{h,n}) \sim \widehat{\sigma}_1} \left[ \left\| \frac{\partial Q^{\pi_\theta}}{\partial a_h} \cdot \frac{\mathrm{d}a_h}{\mathrm{d}\theta} + \frac{\partial Q^{\pi_\theta}}{\partial s_h} \cdot \frac{\mathrm{d}s_h}{\mathrm{d}\theta} - \nabla_\theta \widehat{Q}_t(\widehat{s}_{h,n}, \widehat{a}_{h,n}) \right\|_2 \right]
$$
$$
= \mathbb{E}_{\overline{\sigma}_1, \widehat{\sigma}_1} \left[ \left\| \frac{\partial Q^{\pi_\theta}}{\partial a_h} \cdot \frac{\mathrm{d}a_h}{\mathrm{d}\theta} + \frac{\partial Q^{\pi_\theta}}{\partial s_h} \cdot \frac{\mathrm{d}s_h}{\mathrm{d}\theta} - \frac{\partial \widehat{Q}_t}{\partial \widehat{a}_{h,n}} \cdot \frac{\mathrm{d}\widehat{a}_{h,n}}{\mathrm{d}\theta} - \frac{\partial \widehat{Q}_t}{\partial \widehat{s}_{h,n}} \cdot \frac{\mathrm{d}\widehat{s}_{h,n}}{\mathrm{d}\theta} \right\|_2 \right]
$$
$$
\leq \mathbb{E}_{\overline{\sigma}_1, \widehat{\sigma}_1} \left[ \left\| \frac{\partial Q^{\pi_\theta}}{\partial a_h} \cdot \frac{\mathrm{d}a_h}{\mathrm{d}\theta} - \frac{\partial Q^{\pi_\theta}}{\partial a_h} \cdot \frac{\mathrm{d}\widehat{a}_{h,n}}{\mathrm{d}\theta} \right\|_2 + \left\| \frac{\partial Q^{\pi_\theta}}{\partial a_h} \cdot \frac{\mathrm{d}\widehat{a}_{h,n}}{\mathrm{d}\theta} - \frac{\partial \widehat{Q}_t}{\partial \widehat{a}_{h,n}} \cdot \frac{\mathrm{d}\widehat{a}_{h,n}}{\mathrm{d}\theta} \right\|_2 \right.
$$
$$
\left. + \left\| \frac{\partial Q^{\pi_\theta}}{\partial s_h} \cdot \frac{\mathrm{d}s_h}{\mathrm{d}\theta} - \frac{\partial Q^{\pi_\theta}}{\partial s_h} \cdot \frac{\mathrm{d}\widehat{s}_{h,n}}{\mathrm{d}\theta} \right\|_2 + \left\| \frac{\partial Q^{\pi_\theta}}{\partial s_h} \cdot \frac{\mathrm{d}\widehat{s}_{h,n}}{\mathrm{d}\theta} - \frac{\partial \widehat{Q}_t}{\partial \widehat{s}_{h,n}} \cdot \frac{\mathrm{d}\widehat{s}_{h,n}}{\mathrm{d}\theta} \right\|_2 \right]
$$
$$
\leq L_Q \cdot \left( \mathbb{E}_{\overline{\sigma}_1, \widehat{\sigma}_1} \left[ \left\| \frac{\mathrm{d}a_h}{\mathrm{d}\theta} - \frac{\mathrm{d}\widehat{a}_{h,n}}{\mathrm{d}\theta} \right\|_2 + \left\| \frac{\mathrm{d}s_h}{\mathrm{d}\theta} - \frac{\mathrm{d}\widehat{s}_{h,n}}{\mathrm{d}\theta} \right\|_2 \right] \right) + \left( \frac{\gamma^h}{1-\gamma} \right)^2 \widehat{K}(h) \cdot \epsilon_{v,t}, \quad \text{(B.22)}
$$

where the equality follows from the chain rule and the fact that the critic $\widehat{Q}_t$ has a non-recursive structure, the last inequality follows from (B.21), (B.9) and the definition of $\epsilon_{v,t}$ in (4.5).

Plugging (B.15) and (B.22) into (B.20), we obtain

$$
b_t \leq \kappa\kappa' \cdot h \cdot \left( L_r \cdot \left( \mathbb{E}_{\overline{\sigma}_1, \widehat{\sigma}_1} \left[ \left\| \frac{\mathrm{d}\widehat{s}_{h,n}}{\mathrm{d}\theta} - \frac{\mathrm{d}s_h}{\mathrm{d}\theta} \right\|_2 \right] + \mathbb{E}_{\overline{\sigma}_1, \widehat{\sigma}_1} \left[ \left\| \frac{\mathrm{d}\widehat{a}_{h,n}}{\mathrm{d}\theta} - \frac{\mathrm{d}a_h}{\mathrm{d}\theta} \right\|_2 \right] \right) + 2 L_r \cdot \widehat{K}(h) \right)
$$
$$
+ \kappa' \gamma^h \cdot \left( L_Q \cdot \left( \mathbb{E}_{\overline{\sigma}_1, \widehat{\sigma}_1} \left[ \left\| \frac{\mathrm{d}\widehat{s}_{h,n}}{\mathrm{d}\theta} - \frac{\mathrm{d}s_h}{\mathrm{d}\theta} \right\|_2 \right] + \mathbb{E}_{\overline{\sigma}_1, \widehat{\sigma}_1} \left[ \left\| \frac{\mathrm{d}\widehat{a}_{h,n}}{\mathrm{d}\theta} - \frac{\mathrm{d}a_h}{\mathrm{d}\theta} \right\|_2 \right] \right) + \widehat{K}(h) \cdot \left( \frac{\gamma^h}{1-\gamma} \right)^2 \epsilon_{v,t} \right),
$$
(B.23)

Plugging (B.18), (B.19), and (B.9) into the (B.23), we conclude the proof by obtaining

$$
b_t = O\left( \kappa\kappa' h^2 \frac{1-\gamma^h}{1-\gamma} \widetilde{L}_{\widehat{f}}^h \widetilde{L}_f^h \widetilde{L}_\pi^{2h} \epsilon_{f,t} + \kappa' h \gamma^h \left( \frac{\gamma^h}{1-\gamma} \right)^2 \widetilde{L}_{\widehat{f}}^h \widetilde{L}_\pi^h \epsilon_{v,t} \right). \tag{B.24}
$$

$\square$

**Lemma B.3.** The expected value gradient over the state distribution $\mathbb{P}(s_h)$ can be represented by

$$
\mathbb{E}_{s_h \sim \mathbb{P}(s_h)} \left[ \nabla_\theta V^{\pi_\theta}(s_h) \right] = \mathbb{E}_{(s,a) \sim \overline{\sigma}_1} \left[ \frac{\partial Q^{\pi_\theta}}{\partial a} \cdot \frac{\mathrm{d}a}{\mathrm{d}\theta} + \frac{\partial Q^{\pi_\theta}}{\partial s} \cdot \frac{\mathrm{d}s}{\mathrm{d}\theta} \right],
$$

where $\mathbb{P}(s_h)$ is the state distribution at timestep $h$ when $s_0 \sim \zeta, a_i \sim \pi(\cdot \,|\, s_i)$, and $s_{i+1} \sim f(\cdot \,|\, s_i, a_i)$.

*Proof.* At state $s_h$, the value gradient can be rewritten as

$$\nabla_\theta V^{\pi_\theta}(s_h) = \nabla_\theta \mathbb{E}\left[r(s_h, a_h) + \gamma \cdot \int_{\mathcal{S}} f\big(s_{h+1}|s_h, a_h\big) \cdot V^\pi(s_{h+1}) \mathrm{d}s_{h+1}\right]$$

$$= \nabla_\theta \mathbb{E}\big[r(s_h, a_h)\big] + \gamma \cdot \mathbb{E}\left[\nabla_\theta \int_{\mathcal{S}} f\big(s_{h+1}|s_h, a_h\big) \cdot V^\pi(s_{h+1}) \mathrm{d}s_{h+1}\right]$$

$$= \mathbb{E}\left[\frac{\partial r_h}{\partial a_h} \cdot \frac{\mathrm{d}a_h}{\mathrm{d}\theta} + \frac{\partial r_h}{\partial s_h} \cdot \frac{\mathrm{d}s_h}{\mathrm{d}\theta}\right.$$

$$\left. + \gamma \int_{\mathcal{S}} \Big(\nabla_\theta f\big(s_{h+1}|s_h, a_h\big) \cdot V^\pi(s_{h+1}) + f\big(s_{h+1}|s_h, a_h\big) \cdot \nabla_\theta V^\pi(s_{h+1})\Big) \mathrm{d}s_{h+1}\right]$$

$$= \mathbb{E}\left[\frac{\partial r_h}{\partial a_h} \cdot \frac{\mathrm{d}a_h}{\mathrm{d}\theta} + \frac{\partial r_h}{\partial s_h} \cdot \frac{\mathrm{d}s_h}{\mathrm{d}\theta} + \gamma \int_{\mathcal{S}} \Big(\nabla_a f(s_{h+1}|s_h, a) \cdot \frac{\mathrm{d}a_h}{\mathrm{d}\theta} \cdot V^\pi(s_{h+1})\right.$$

$$\left. + \nabla_s f(s_{h+1}|s_h, a_h) \cdot \frac{\mathrm{d}s_h}{\mathrm{d}\theta} \cdot V^\pi(s_{h+1}) + f\big(s_{h+1}|s_h, a_h\big) \cdot \nabla_\theta V^\pi(s_{h+1})\Big) \mathrm{d}s_{h+1}\right],$$
(B.25)

where the first equation follows from the Bellman equation and the last two equations hold due to the chain rule. Here, it is worth noting that when $h \geq 1$, both $a_h$ and $s_h$ have dependencies on all previous timesteps. For any $h \geq 1$, we have from the chain rule that $\nabla_\theta r(s_h, a_h) = \partial r_h / \partial a_h \cdot \mathrm{d}a_h/\mathrm{d}\theta + \partial r_h/\partial s_h \cdot \mathrm{d}s_h/\mathrm{d}\theta$. This differs from the case when $h = 0$, e.g., in the deterministic policy gradient theorem [52], where we can simply write $\nabla_\theta r(s_h, a_h) = \partial r_h/\partial a_h \cdot \partial a_h/\partial \theta$.

Rearranging terms in (B.25) gives

$$\nabla_\theta V^{\pi_\theta}(s_h) = \mathbb{E}\left[\nabla_a\left(r(s_h, a_h) + \gamma \int_{\mathcal{S}} f(s_{h+1}|s_h, a_h) \cdot V^\pi(s_{h+1}) \mathrm{d}s_{h+1}\right) \cdot \frac{\mathrm{d}a_h}{\mathrm{d}\theta}\right.$$

$$+ \nabla_s\left(r(s_h, a_h) + \gamma \int_{\mathcal{S}} f(s_{h+1}|s_h, a_h) \cdot V^\pi(s_{h+1}) \mathrm{d}s_{h+1}\right) \cdot \frac{\mathrm{d}s_h}{\mathrm{d}\theta}$$

$$\left. + \gamma \int_{\mathcal{S}} f\big(s_{h+1}|s_h, a_h\big) \cdot \nabla_\theta V^\pi(s_{h+1}) \mathrm{d}s_{h+1}\right]$$

$$= \mathbb{E}\left[\frac{\partial Q^{\pi_\theta}}{\partial a_h} \cdot \frac{\mathrm{d}a_h}{\mathrm{d}\theta} + \frac{\partial Q^{\pi_\theta}}{\partial s_h} \cdot \frac{\mathrm{d}s_h}{\mathrm{d}\theta} + \gamma \int_{\mathcal{S}} f\big(s_{h+1}|s_h, a_h\big) \cdot \nabla_\theta V^\pi(s_{h+1}) \mathrm{d}s_{h+1}\right],$$
(B.26)

where the last equation holds since $Q^{\pi_\theta}(s_h, a_h) = r(s_h, a_h) + \gamma \int_{\mathcal{S}} f(s_{h+1}|s_h, a_h) \cdot V^\pi(s_{h+1}) \mathrm{d}s_{h+1}$.

By recursively applying (B.26), we obtain

$$\nabla_\theta V^{\pi_\theta}(s_h) = \mathbb{E}\left[\int_{\mathcal{S}} \sum_{i=h}^{\infty} \gamma^{i-h} \cdot f\big(s_{i+1}|s_i, a_i\big) \cdot \left(\frac{\partial Q^{\pi_\theta}}{\partial a_i} \cdot \frac{\mathrm{d}a_i}{\mathrm{d}\theta} + \frac{\partial Q^{\pi_\theta}}{\partial s_i} \cdot \frac{\mathrm{d}s_i}{\mathrm{d}\theta}\right) \mathrm{d}s_{i+1}\right]. \quad \text{(B.27)}$$

Let $\overline{\sigma}_2(s, a) = (1 - \gamma) \cdot \sum_{i=h}^{\infty} \gamma^{i-h} \cdot \mathbb{P}(s_i = s, a_i = a)$, where $s_0 \sim \zeta$, $a_i \sim \pi(\cdot \,|\, s_i)$, and $s_{i+1} \sim f(\cdot \,|\, s_i, a_i)$. By definition we have

$$\sigma(s, a) = (1 - \gamma) \cdot \sum_{i=0}^{h-1} \gamma^i \cdot \mathbb{P}(s_i = s, a_i = a) + \gamma^h \cdot \overline{\sigma}_1(s, a)$$

$$= (1 - \gamma) \cdot \sum_{i=0}^{h-1} \gamma^i \cdot \mathbb{P}(s_i = s, a_i = a) + \gamma^h \cdot \overline{\sigma}_2(s, a).$$

Therefore we have the equivalence $\overline{\sigma}_1(s, a) = \overline{\sigma}_2(s, a)$.

By taking the expectation over $s_h$ in (B.27), we have the stated result, i.e.,

$$\mathbb{E}_{s_h \sim \mathbb{P}(s_h)}\big[\nabla_\theta V^{\pi_\theta}(s_h)\big] = \mathbb{E}_{(s,a) \sim \overline{\sigma}_2}\left[\frac{\partial Q^{\pi_\theta}}{\partial a} \cdot \frac{\mathrm{d}a}{\mathrm{d}\theta} + \frac{\partial Q^{\pi_\theta}}{\partial s} \cdot \frac{\mathrm{d}s}{\mathrm{d}\theta}\right] = \mathbb{E}_{(s,a) \sim \overline{\sigma}_1}\left[\frac{\partial Q^{\pi_\theta}}{\partial a} \cdot \frac{\mathrm{d}a}{\mathrm{d}\theta} + \frac{\partial Q^{\pi_\theta}}{\partial s} \cdot \frac{\mathrm{d}s}{\mathrm{d}\theta}\right].$$

$$\square$$

**Lemma B.4.** Recall that the state distribution $\mu_\pi$ where $\widehat{s}_{0,n}$ is sampled from is of the form $\mu_\pi(s) = \beta \cdot \nu_\pi(s) + (1 - \beta) \cdot \zeta(s)$. The gradient bias $b_t$ at any iteration $t$ satisfies

$$
b_t \leq \kappa\big(\beta + \kappa \cdot (1 - \beta)\big) \cdot \mathbb{E}_{s_0 \sim \nu_\pi, \widehat{s}_{0,n} \sim \nu_\pi}\left[\left\|\nabla_\theta \sum_{i=0}^{h-1} \gamma^i \cdot r(s_i, a_i) - \nabla_\theta \sum_{i=0}^{h-1} \gamma^i \cdot r(\widehat{s}_{i,n}, \widehat{a}_{i,n})\right\|_2\right]
$$
$$
+ \big(\beta + \kappa \cdot (1 - \beta)\big)\gamma^h \cdot
$$
$$
\mathbb{E}_{(s_h, a_h) \sim \overline{\sigma}_1, (\widehat{s}_{h,n}, \widehat{a}_{h,n}) \sim \widehat{\sigma}_1}\left[\left\|\frac{\partial Q^{\pi_\theta}}{\partial a_h} \cdot \frac{\mathrm{d}a_h}{\mathrm{d}\theta} + \frac{\partial Q^{\pi_\theta}}{\partial s_h} \cdot \frac{\mathrm{d}s_h}{\mathrm{d}\theta} - \nabla_\theta \widehat{Q}_t(\widehat{s}_{h,n}, \widehat{a}_{h,n})\right\|_2\right].
$$

*Proof.* To begin, we decompose the gradient bias by

$$
b_t = \left\|\nabla_\theta J(\pi_{\theta_t}) - \mathbb{E}\big[\widehat{\nabla}_\theta J(\pi_{\theta_t})\big]\right\|_2
$$
$$
= \left\|\mathbb{E}\big[\nabla_\theta J(\pi_{\theta_t}) - \widehat{\nabla}_\theta J(\pi_{\theta_t})\big]\right\|_2 \tag{B.28}
$$
$$
= \left\|\mathbb{E}_{s_0 \sim \zeta, \widehat{s}_{0,n} \sim \mu_\pi}\left[\nabla_\theta \sum_{i=0}^{h-1} \gamma^i \cdot r(s_i, a_i) + \gamma^h \cdot \nabla_\theta V^{\pi_\theta}(s_h) - \nabla_\theta \sum_{i=0}^{h-1} \gamma^i \cdot r(\widehat{s}_{i,n}, \widehat{a}_{i,n})\right.\right.
$$
$$
\left.\left. - \gamma^h \cdot \nabla_\theta \widehat{V}_t(\widehat{s}_{h,n})\right]\right\|_2,
$$

where we note that $s_0$ and $\widehat{s}_{0,n}$ are sampled from $\zeta$ and $\mu_\pi$ following the definition of the RL objective and the form of gradient estimator, respectively.

For $\mu_\pi(s) = \beta \cdot \nu_\pi(s) + (1 - \beta) \cdot \zeta(s)$, let $Z$ be the random variable satisfying $\mathbb{P}(Z = 0) = \beta$ and $\mathbb{P}(Z = 1) = 1 - \beta$, i.e., the event $Z = 0$ and $Z = 1$ corresponds to that the state $s$ is sampled from $\nu_\pi$ and $\zeta$, respectively. For any random variable $Y$, following the law of total expectation, we know that

$$
\mathbb{E}_{\mu_\pi}[Y] = \mathbb{E}[\mathbb{E}[Y|Z]] = \mathbb{E}[Y|Z = 0]\mathbb{P}(Z = 0) + \mathbb{E}[Y|Z = 1]\mathbb{P}(Z = 1)
$$
$$
= \beta\mathbb{E}[Y|Z = 0] + (1 - \beta)\mathbb{E}[Y|Z = 1]
$$
$$
= \beta\mathbb{E}_{\nu_\pi}[Y] + (1 - \beta)\mathbb{E}_\zeta[Y]. \tag{B.29}
$$

Therefore, we have from (B.28) that

$$
b_t \leq \mathbb{E}_{\widehat{s}_{0,n} \sim \mu_\pi}\left[\left\|\mathbb{E}_{s_0 \sim \zeta}\left[\nabla_\theta \sum_{i=0}^{h-1} \gamma^i \cdot r(s_i, a_i) + \gamma^h \cdot \nabla_\theta V^{\pi_\theta}(s_h) - \nabla_\theta \sum_{i=0}^{h-1} \gamma^i \cdot r(\widehat{s}_{i,n}, \widehat{a}_{i,n})\right.\right.\right.
$$
$$
\left.\left.\left. - \gamma^h \cdot \nabla_\theta \widehat{V}_t(\widehat{s}_{h,n})\right]\right\|_2\right]
$$
$$
\leq \beta\mathbb{E}_{\widehat{s}_{0,n} \sim \nu_\pi}\left[\left\|\mathbb{E}_{s_0 \sim \zeta}\left[\nabla_\theta \sum_{i=0}^{h-1} \gamma^i \cdot r(s_i, a_i) + \gamma^h \cdot \nabla_\theta V^{\pi_\theta}(s_h) - \nabla_\theta \sum_{i=0}^{h-1} \gamma^i \cdot r(\widehat{s}_{i,n}, \widehat{a}_{i,n})\right.\right.\right.
$$
$$
\left.\left.\left. - \gamma^h \cdot \nabla_\theta \widehat{V}_t(\widehat{s}_{h,n})\right]\right\|_2\right] + (1 - \beta)\mathbb{E}_{\widehat{s}_{0,n} \sim \zeta}\left[\left\|\mathbb{E}_{s_0 \sim \zeta}\left[\nabla_\theta \sum_{i=0}^{h-1} \gamma^i \cdot r(s_i, a_i) + \gamma^h \cdot \nabla_\theta V^{\pi_\theta}(s_h)\right.\right.\right.
$$
$$
\left.\left.\left. - \nabla_\theta \sum_{i=0}^{h-1} \gamma^i \cdot r(\widehat{s}_{i,n}, \widehat{a}_{i,n}) - \gamma^h \cdot \nabla_\theta \widehat{V}_t(\widehat{s}_{h,n})\right]\right\|_2\right], \tag{B.30}
$$

where the first inequality holds since $\|\mathbb{E}[\cdot]\|_2 \leq \mathbb{E}[\|\cdot\|_2]$ and the second inequality holds due to (B.29).

Using the result from Lemma B.3, we know that

$$\mathbb{E}_{s_0\sim\zeta}\left[\nabla_\theta\sum_{i=0}^{h-1}\gamma^i\cdot r(s_i,a_i)+\gamma^h\cdot\nabla_\theta V^{\pi_\theta}(s_h)-\nabla_\theta\sum_{i=0}^{h-1}\gamma^i\cdot r(\widehat{s}_{i,n},\widehat{a}_{i,n})-\gamma^h\cdot\nabla_\theta\widehat{V}_t(\widehat{s}_{h,n})\right]$$

$$=\underbrace{\mathbb{E}_{s_0\sim\zeta}\left[\nabla_\theta\sum_{i=0}^{h-1}\gamma^i\cdot r(s_i,a_i)-\nabla_\theta\sum_{i=0}^{h-1}\gamma^i\cdot r(\widehat{s}_{i,n},\widehat{a}_{i,n})\right]}_{B_r}$$

$$+\underbrace{\gamma^h\mathbb{E}_{(s_h,a_h)\sim\overline{\sigma}_1}\left[\frac{\partial Q^{\pi_\theta}}{\partial a_h}\cdot\frac{\mathrm{d}a_h}{\mathrm{d}\theta}+\frac{\partial Q^{\pi_\theta}}{\partial s_h}\cdot\frac{\mathrm{d}s_h}{\mathrm{d}\theta}-\nabla_\theta\widehat{V}_t(\widehat{s}_{h,n})\right]}_{B_v}.$$

Here, the shorthand notation $B_r$ denotes the bias introduced by the $h$-step model expansion and $B_v$ denotes the bias introduced by using a critic for tail estimation. Then we may rewrite (B.30) as

$$b_t\le\beta\cdot\mathbb{E}_{\widehat{s}_{0,n}\sim\nu_\pi}\left[\left\|B_r+B_v\right\|_2\right]+(1-\beta)\cdot\mathbb{E}_{\widehat{s}_{0,n}\sim\zeta}\left[\left\|B_r+B_v\right\|_2\right]$$

$$\le\left(\beta\cdot\mathbb{E}_{\widehat{s}_{0,n}\sim\nu_\pi}\left[\left\|B_r\right\|_2\right]+(1-\beta)\cdot\mathbb{E}_{\widehat{s}_{0,n}\sim\zeta}\left[\left\|B_r\right\|_2\right]\right)$$

$$+\left(\beta\cdot\mathbb{E}_{\widehat{s}_{0,n}\sim\nu_\pi}\left[\left\|B_v\right\|_2\right]+(1-\beta)\cdot\mathbb{E}_{\widehat{s}_{0,n}\sim\zeta}\left[\left\|B_v\right\|_2\right]\right). \qquad\text{(B.31)}$$

For the first term on the right-hand side of (B.31), we have

$$\beta\cdot\mathbb{E}_{\widehat{s}_{0,n}\sim\nu_\pi}\left[\left\|B_r\right\|_2\right]+(1-\beta)\cdot\mathbb{E}_{\widehat{s}_{0,n}\sim\zeta}\left[\left\|B_r\right\|_2\right]$$

$$=\beta\cdot\mathbb{E}_{\widehat{s}_{0,n}\sim\nu_\pi}\left[\left\|\mathbb{E}_{s_0\sim\zeta}\left[\nabla_\theta\sum_{i=0}^{h-1}\gamma^i\cdot r(s_i,a_i)-\nabla_\theta\sum_{i=0}^{h-1}\gamma^i\cdot r(\widehat{s}_{i,n},\widehat{a}_{i,n})\right]\right\|_2\right]+(1-\beta)\cdot$$

$$\mathbb{E}_{\widehat{s}_{0,n}\sim\nu_\pi}\left[\left\|\mathbb{E}_{s_0\sim\zeta}\left[\nabla_\theta\sum_{i=0}^{h-1}\gamma^i\cdot r(s_i,a_i)-\nabla_\theta\sum_{i=0}^{h-1}\gamma^i\cdot r(\widehat{s}_{i,n},\widehat{a}_{i,n})\right]\right\|_2\right]\cdot\left\{\mathbb{E}_{\nu_\pi}\left[\left(\frac{\mathrm{d}\zeta}{\mathrm{d}\nu_\pi}(s)\right)^2\right]\right\}^{1/2}$$

$$\le\big(\beta+\kappa\cdot(1-\beta)\big)\cdot\mathbb{E}_{\widehat{s}_{0,n}\sim\nu_\pi}\left[\left\|\mathbb{E}_{s_0\sim\zeta}\left[\nabla_\theta\sum_{i=0}^{h-1}\gamma^i\cdot r(s_i,a_i)-\nabla_\theta\sum_{i=0}^{h-1}\gamma^i\cdot r(\widehat{s}_{i,n},\widehat{a}_{i,n})\right]\right\|_2\right]$$

$$\le\kappa\big(\beta+\kappa\cdot(1-\beta)\big)\cdot\mathbb{E}_{s_0\sim\nu_\pi,\widehat{s}_{0,n}\sim\nu_\pi}\left[\left\|\nabla_\theta\sum_{i=0}^{h-1}\gamma^i\cdot r(s_i,a_i)-\nabla_\theta\sum_{i=0}^{h-1}\gamma^i\cdot r(\widehat{s}_{i,n},\widehat{a}_{i,n})\right\|_2\right],$$

$$\text{(B.32)}$$

where the first and second inequalities follow from the definition of $\kappa$ in Proposition 5.6.

Similarly, for the second term on the right-hand side of (B.31), we have

$$\beta\cdot\mathbb{E}_{\widehat{s}_{0,n}\sim\nu_\pi}\left[\left\|B_v\right\|_2\right]+(1-\beta)\cdot\mathbb{E}_{\widehat{s}_{0,n}\sim\zeta}\left[\left\|B_v\right\|_2\right]$$

$$=\beta\cdot\mathbb{E}_{\widehat{s}_{0,n}\sim\nu_\pi}\left[\gamma^h\cdot\left\|\mathbb{E}_{(s_h,a_h)\sim\overline{\sigma}_1}\left[\frac{\partial Q^{\pi_\theta}}{\partial a_h}\cdot\frac{\mathrm{d}a_h}{\mathrm{d}\theta}+\frac{\partial Q^{\pi_\theta}}{\partial s_h}\cdot\frac{\mathrm{d}s_h}{\mathrm{d}\theta}-\nabla_\theta\widehat{V}_t(\widehat{s}_{h,n})\right]\right\|_2\right]$$

$$+(1-\beta)\cdot\mathbb{E}_{\widehat{s}_{0,n}\sim\zeta}\left[\gamma^h\cdot\left\|\mathbb{E}_{(s_h,a_h)\sim\overline{\sigma}_1}\left[\frac{\partial Q^{\pi_\theta}}{\partial a_h}\cdot\frac{\mathrm{d}a_h}{\mathrm{d}\theta}+\frac{\partial Q^{\pi_\theta}}{\partial s_h}\cdot\frac{\mathrm{d}s_h}{\mathrm{d}\theta}-\nabla_\theta\widehat{V}_t(\widehat{s}_{h,n})\right]\right\|_2\right]$$

$$\le\big(\beta+\kappa\cdot(1-\beta)\big)\gamma^h\cdot\mathbb{E}_{(s_h,a_h)\sim\overline{\sigma}_1,\widehat{s}_{0,n}\sim\nu_\pi}\left[\left\|\frac{\partial Q^{\pi_\theta}}{\partial a_h}\cdot\frac{\mathrm{d}a_h}{\mathrm{d}\theta}+\frac{\partial Q^{\pi_\theta}}{\partial s_h}\cdot\frac{\mathrm{d}s_h}{\mathrm{d}\theta}-\nabla_\theta\widehat{V}_t(\widehat{s}_{h,n})\right\|_2\right]$$

$$=\big(\beta+\kappa\cdot(1-\beta)\big)\gamma^h\cdot$$

$$\mathbb{E}_{(s_h,a_h)\sim\overline{\sigma}_1,(\widehat{s}_{h,n},\widehat{a}_{h,n})\sim\widehat{\sigma}_1}\left[\left\|\frac{\partial Q^{\pi_\theta}}{\partial a_h}\cdot\frac{\mathrm{d}a_h}{\mathrm{d}\theta}+\frac{\partial Q^{\pi_\theta}}{\partial s_h}\cdot\frac{\mathrm{d}s_h}{\mathrm{d}\theta}-\nabla_\theta\widehat{Q}_t(\widehat{s}_{h,n},\widehat{a}_{h,n})\right\|_2\right]. \quad\text{(B.33)}$$

Plugging (B.32) and (B.33) into (B.31) completes the proof. $\qquad\square$

**Lemma B.5** (Lipschitz Value Function [47] Theorem 1). *Under Assumption 5.3, for $\gamma L_f(1 + L_\pi) < 1$, then the state-action value function is $L_Q$-Lipschitz continuous, such that for any policy $\pi_\theta$, state $s_1, s_2 \in \mathcal{S}$ and action $a_1, a_2 \in \mathcal{A}$, $\left| Q^{\pi_\theta}(s_1, a_1) - Q^{\pi_\theta}(s_2, a_2) \right| \leq L_Q \cdot \left\| (s_1 - s_2, a_1 - a_2) \right\|_2$, and*

$$L_Q = L_r/(1 - \gamma L_f(1 + L_\pi)).$$

## B.4 Proof of Proposition 5.7

*Proof.* When $\gamma \approx 1$, we have

$$\frac{1 - \gamma^h}{1 - \gamma} = \sum_{i=0}^{h-1} \gamma^i \approx h, \quad \frac{\gamma^h}{1 - \gamma} = \frac{1}{1 - \gamma} - \frac{1 - \gamma^h}{1 - \gamma} \approx \frac{1}{1 - \gamma} - h.$$

We denote by $H = 1/(1 - \gamma) = \sum_{i=0}^{\infty} \gamma^i$ the effective task horizon.

To find the optimal unroll length $h^*$ that minimizes the upper bound of the convergence, we define $g(h)$ as follows,

$$g(h) = c \cdot \left(2\delta \cdot b'_t + \frac{\eta}{2} \cdot v_t'^2\right) + b_t'^2 + v_t'^2.$$

Here, $v_t'^2$ and $b'_t$ are the leading terms in the variance, bias bound (i.e., (B.12) and (B.24)) when $L_f, L_{\widehat{f}}$, and $L_\pi$ are less than or equal to 1. Formally, $v'_t = h^3$ and $b'_t = h^3 \epsilon_{f,t} + h(H - h)^2 \epsilon_{v,t}$. We consider the terms that are only dependent on $h$, $H$, $\epsilon_{f,t}$, and $\epsilon_{v,t}$ to simplify the analysis and determine the order of $h^*$.

Our first problem is to find the optimal model unroll $h'^*$ that minimizes $g(h)$. We notice that $g(h)$ increases monotonically with respect to $b'_t$ and $v'_t$ when they are non-negative. This further simplifies the problem to find

$$h'^* = \underset{h}{\arg\min}\, b'_t + c'v'_t = \underset{h}{\arg\min}\, \underbrace{h^3(\epsilon_{f,t} + c') + h(H - h)^2 \epsilon_{v,t}}_{g_1(h)}, \tag{B.34}$$

where $c'$ is some constant that does not affect the order of $h'^*$.

By taking the derivative of the right-hand side of (B.34) with respect to $h$ and setting it to zero, we obtain

$$\frac{\partial}{\partial h} g_1(h) = 3h^2 \cdot (\epsilon_{f,t} + c') + (3h^2 - 4Hh + H^2) \cdot \epsilon_{v,t} = 0. \tag{B.35}$$

Solve the above quadratic equation with respect to $h$, we have the two non-negative roots $h_1'^*$ and $h_2'^*$ as follows,

$$h_1'^* = \frac{4H\epsilon_{v,t} + \sqrt{(4H\epsilon_{v,t})^2 - 12c_1\epsilon_{v,t}H^2}}{6c_1}, \quad h_2'^* = \frac{4H\epsilon_{v,t} - \sqrt{(4H\epsilon_{v,t})^2 - 12c_1\epsilon_{v,t}H^2}}{6c_1},$$

where we define $c_1 = \epsilon_{f,t} + \epsilon_{v,t} + c'$.

Now we study the resulting two cases. If $(4H\epsilon_{v,t})^2 - 12c_1\epsilon_{v,t}H^2 \geq 0$, we have

$$h'^* = h_1'^* = O\big(\epsilon_{v,t}/(\epsilon_{f,t} + \epsilon_{v,t}) \cdot H\big).$$

We can verify that $h_1'^*$ is indeed the minimum by calculating the second-order derivative at $h_1'^*$ as follows,

$$\frac{\partial^2 g_1(h_1'^*)}{\partial h^2} = \frac{4H\epsilon_{v,t} + \sqrt{(4H\epsilon_{v,t})^2 - 4c_1 \cdot H^2}}{6c_1} * 6(\epsilon_{f,t} + \epsilon_{v,t} + c') - 4H\epsilon_{v,t}$$

$$= \sqrt{(4H\epsilon_{v,t})^2 - 4c_1 \cdot H^2} > 0.$$

The other case is $(4H\epsilon_{v,t})^2 - 12c_1\epsilon_{v,t}H^2 < 0$. When this happens, (B.35) does not have a real solution $h'^*$ and we set $h^*$ to 0. This concludes the proof of Proposition 5.7. □

## B.5 Proof of Corollary 5.9

*Proof.* We let the learning rate $\eta = 1/\sqrt{T}$. Then for $T \geq 4L^2$, we have $c = (\eta - L\eta^2)^{-1} \leq 2\sqrt{T}$ and $L\eta \leq 1/2$. By setting $N = O(\sqrt{T})$, we obtain

$$
\begin{aligned}
\min_{t \in [T]} \mathbb{E}\Big[\big\|\nabla_\theta J(\pi_{\theta_t})\big\|_2^2\Big] &\leq \frac{4}{T} \cdot \left(\sum_{t=0}^{T-1} c \cdot \left(2\delta \cdot b_t + \frac{\eta}{2} \cdot v_t\right) + b_t^2 + v_t\right) + \frac{4c}{T} \cdot \mathbb{E}\big[J(\pi_{\theta_T}) - J(\pi_{\theta_1})\big] \\
&\leq \frac{4}{T}\left(\sum_{t=0}^{T-1} 4\sqrt{T}\delta \cdot b_t + b_t^2 + 2v_t\right) + \frac{8}{\sqrt{T}} \cdot \mathbb{E}\big[J(\pi_{\theta_T}) - J(\pi_{\theta_1})\big] \\
&\leq \frac{4}{T}\left(\sum_{t=0}^{T-1} 4\sqrt{T}\delta \cdot b_t + b_t^2\right) + O\big(1/\sqrt{T}\big) \\
&\leq \frac{16\delta}{\sqrt{T}}\varepsilon(T) + \frac{4}{T}\varepsilon^2(T) + O\big(1/\sqrt{T}\big).
\end{aligned}
$$

This concludes the proof. $\qquad\square$

# C   Experimental Details

## C.1   Implementations and Comparisons with More RL Baselines

For the model-based baseline Model-Based Policy Optimization (MBPO) [28], we use the implementation in the Mbrl-lib [45]. For all other model-free baselines, we use the implementations in Tianshou [62] that have state-of-the-art results.

We observe that the RP-DP has competitive performance in all the evaluation tasks compared to the popular baselines, suggesting the importance of studying model-based RP PGMs. In experiments, we implement RP-DR as the on-policy SVG(1) [25]. We observe that the training can be unstable when using the off-policy SVG implementation, which requires a carefully chosen policy update rate as well as a proper size of the experience replay buffer. This is because when the learning rate is large, the magnitude of the inferred policy noise (from the previous data samples in the experience replay) can be huge. Implementing an on-policy version of RP-DR can avoid such an issue, following [25]. This, however, can degrade the performance of RP-DR compared to the off-policy RP-DP algorithm in several tasks. We conjecture that implementing the off-policy version of RP-DR can boost its performance, which requires techniques to stabilize training and we leave it as future work. For RP-DP, we implement it as Model-Augmented Actor-Critic (MAAC) [12] with entropy regularization [23], as suggested by [4]. RP(0) represents setting $h = 0$ in the RP PGM formulas [4], which is a model-free algorithm that is a stochastic counterpart of deterministic policy gradients.

For model-free baselines, we compare with Likelihood Ratio (LR) policy gradient methods (c.f. (2.2)), including REINFORCE [56], Natural Policy Gradient (NPG) [31], Advantage Actor Critic (A2C), Actor Critic using Kronecker-Factored Trust Region (ACKTR) [65], and Proximal Policy Optimization (PPO) [49]. We also evaluate algorithms that are built upon DDPG [34], including Soft Actor-Critic (SAC) [23] and Twin Delayed Deep Deterministic policy gradient (TD3) [19].

## C.2   Implementation of Spectral Normalization

In experiments, we use Multilayer Perceptrons (MLPs) for the critic, policy, and model. Besides, we adopt Gaussian dynamical models and policies as the source of stochasticity. To test the benefit of smooth function approximations in model-based RP policy gradient algorithms, spectral normalization is applied to all layers of the policy MLP and all except the final layers of the model MLP. The number of layers for the policy and the dynamics model is $4$ and $5$, respectively.

Our code is based on PyTorch [44], which has an out-of-the-shelf implementation of spectral normalization. Thus, applying SN to the MLP is pretty simple and no additional lines of code are needed. Specifically, we only need to import and apply SN to each layer:

```
from torch.nn.utils.parametrizations import spectral_norm
layer = [spectral_norm(nn.Linear(in_dim, hidden_dim)), nn.ReLU()]
```

## C.3 Ablation on Spectral Normalization

In this section, we conduct ablation studies on the spectral normalization applied to model-based RP PGMs. Specifically, we aim to answer the following two questions: (1) What are the effects of SN when applied to different NN components of model-based RP PGMs? (2) Does SN improve other MBRL algorithms by smoothing the models?

**What are the effects of SN when applied to different NN components of model-based RP PGMs?**
We study the following NN components that spectral normalization is applied to: both the model and the policy (default setting as suggested by our theory); only the model; only the policy; no SN is applied (vanilla setting). The results are shown in Figure 8.

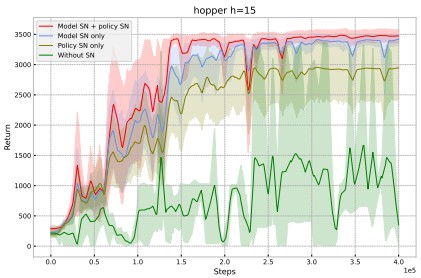 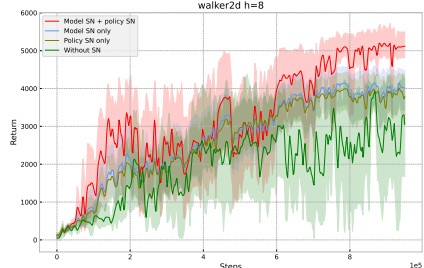

Figure 8: Ablation on the effects of spectral normalization when applied to different NN components of model-based RP PGMs.

We observe in Fig. 8 that for both the hopper and the walker2d tasks, applying SN to the model and policy simultaneously achieves the best performance, which supports our theoretical results. Besides, learning a smooth transition kernel by applying SN to the neural network model only is slightly better than only applying SN to the policy. At the same time, the vanilla implementation of model-based RP PGM fails to give acceptable results.

**Does SN improve other MBRL algorithms by smoothing the models?** We have established that smoothness regularization, such as SN, in model-based RP PGMs can reduce the gradient variance and improve their convergence and performance. However, it is not necessarily the case for other model-based RL methods. In this part, we investigate whether SN can improve previous MBRL algorithms due to a smoothed model. Specifically, we evaluate MBPO [28], a popular MBRL algorithm when SN is added to different numbers of layers in the model neural network. The results are shown in Figure 9. We observe that SN has a negative influence when applied, which is in contrast to our findings that SN is beneficial to RP PGMs.

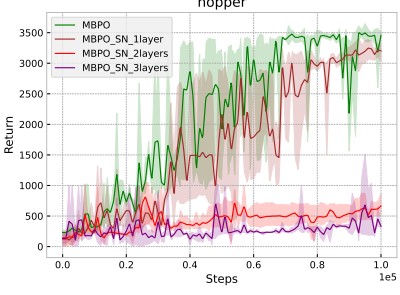 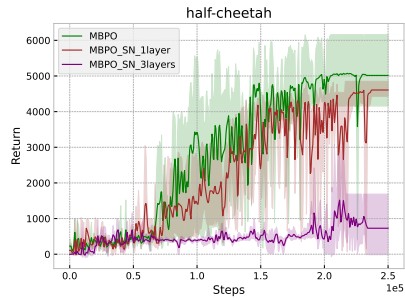

Figure 9: Ablation on the effect of SN in other MBRL algorithms. Spectral normalization has a negative effect when applied to MBPO, indicating that SN by smoothing the model, does *not* necessarily lead to better gradient estimation or improve the performance for MBRL methods other than model-based RP PGMs.

## C.4 Ablation on Different Model Learners

Our main theoretical results in Section 5 depend on the model error defined in (4.4), which, however, cannot directly serve as the model training objective. For this reason, we evaluate different model learners: single- and multi-step ($h$-step) state prediction models, as well as multi-step predictive models integrated with the directional derivative error [33]. The results are reported in Figure 10. We observe that enlarging the prediction steps benefits training. The algorithm also converges faster in walker2d when considering derivative error, which approximately minimizes (4.4) and supports our analysis. However, calculating the directional derivative error by searching $k$ nearest points in the buffer significantly increases the computational cost, for which reason we use $h$-step state predictive models as default in experiments.

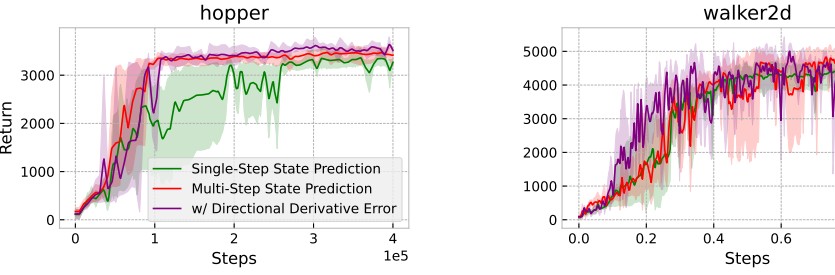

Figure 10: Ablation on different model learners: single-step and multi-step state prediction models, and multi-step state prediction models trained with an additional directional derivative error.

## C.5 Figures in the Main Text in Larger Sizes

Here, we provide identical figures that are larger in size. Figure 11, 12, 13, 14 correspond to Figure 1, 4, 6, 7 in the main text, respectively.

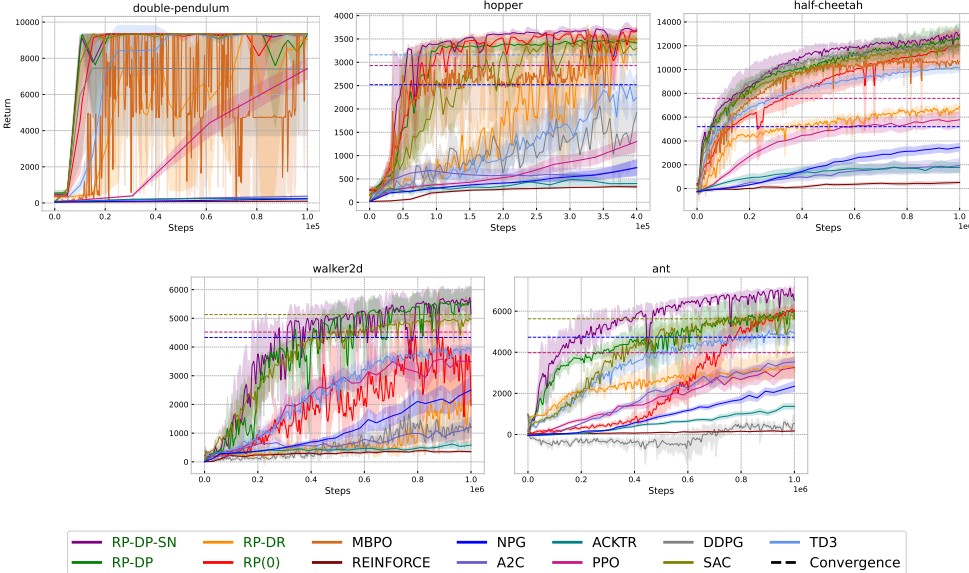

Figure 11: Comparisons between RP PGMs (the green labels) and MF/MB baselines (the black labels) in the MuJoCo [57] tasks.

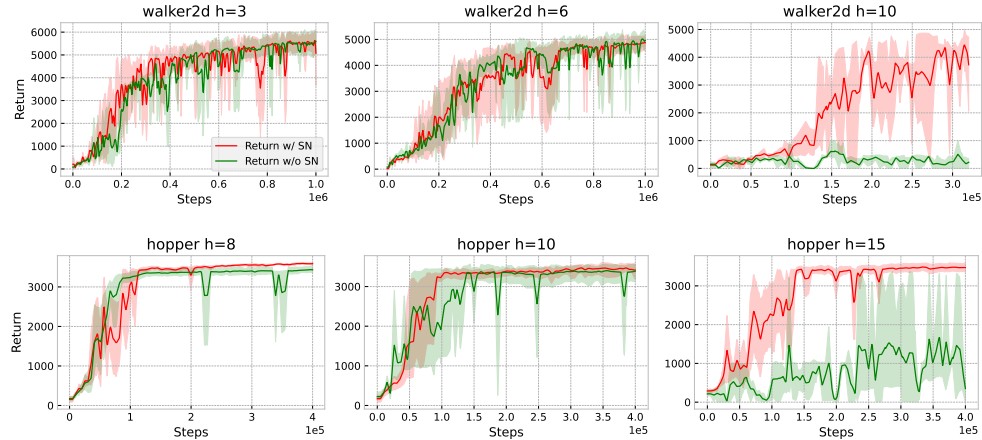

Figure 12: Performance of vanilla and SN-based MB RP PGMs with varying $h$. The vanilla method only works with a small $h$ and fails when $h$ increases, while the SN-based method enables a larger $h$.

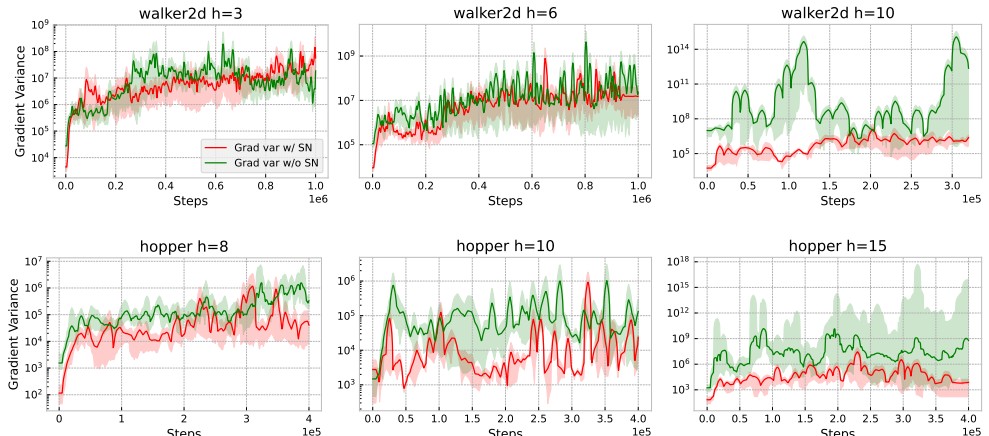

Figure 13: Gradient variance of RP PGMs. The variance is significantly lower with SN when $h$ is large.

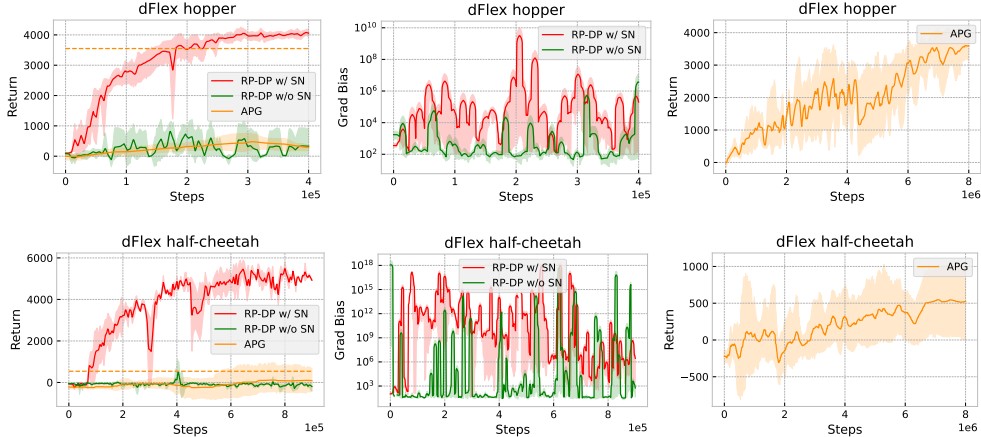

Figure 14: Performance and gradient bias in differentiable simulation. The last column is the full training curves of APG, which need 20 times more steps than RP-DP-SN to reach a comparable return in the hopper task and fail in the half-cheetah task, respectively.

