# OpenReview forum: "Model-Based Reparameterization Policy Gradient Methods: Theory and Practical Algorithms"
_NeurIPS.cc/2023/Conference — NeurIPS 2023 poster_

### Official Review · Reviewer_iodf · 2023-06-15

**Soundness:** 3 good
**Presentation:** 3 good
**Contribution:** 3 good
**Rating:** 6
**Confidence:** 3

**Summary:**

When using re-parameterization (RP) gradient estimators for policy gradient methods (PGM) the
optimisation landscape becomes chaotic and non-smooth, with exploding gradient variance.
This paper examines RP gradient estimators for policy gradient methods.
First, the authors theoretically examine RP PGM.
Their theoretical examination highlights that the smoothness of the dynamic model's and the policy's function approximators has a large impact on the quality of the gradient estimator.
Following this theoretically guided insight, they propose to enforce smoothness by applying spectral normalisation to all layers of the dynamics and policy's neural networks.
Their results suggest that this simple modification mitigates the exploding gradient issue associated with RP PGM.
Overall, I think this is a nice paper as it presents a practically significant result backed by theory.


**Strengths:**

The paper provides an interesting insight into how the smoothness of function approximators influences the performance of reparameterisation gradients.
In particular, the paper builds a practical algorithm on top of theoretical insights and I think this is its greatest strength; as it provides a strong contribution to the community.
The paper's contributions are clear and the work is positioned well against the literature.
The theoretical claims appear sound but some of the derivations in the appendix are out of the scope of my expertise.

**Weaknesses:**

The paper's biggest weakness is its clarity. First of all, the figures are of very low quality and in some cases they are illegible.
There are also no figures until the results section on page 8. I think the paper would benefit by moving some of the figures earlier in the paper and making them bigger.
I also have concerns regarding reproducibility as no hyperparameters are reported in the paper/appendix.

In general, the figures are of very low quality.
- Figure 1
  - I can't read the title or axis labels. These should be made bigger.
  - The figure is also very crowded.
    - I cannot see the variance for several algorithms.
    - Consider running more random seeds to smooth the curves.
    - Consider having fewer plots in a row so that they can be bigger. Put the rest in the appendix.
- Figures 4/6/7
  - I can't read the title or axis labels. These should be made bigger.
  - Do you need both the walker2d and hopper figures in the main paper? The figures would be easier to read if they were bigger, for example, if there were only three plots per row. Perhaps the hopper results could be moved to the appendix?
- What does the shading represent? variance/std? How many seeds were used in the experiments to calculate the variance?
- In Figure 7 the authors refer to the third/last columns. Consider labelling the columns a-f.

Reproducibility
  - How many seeds were used for each experiment?
  - What hyperparameters were used? Learning rates, optimiser, activation functions, batch size, width of hidden layers, number of epochs, discount factor, early stopping callback, etc. I cannot see any of these details in the paper or appendix

Some citations use arXiv versions instead of conference publications.
 - 11 is published at ICLR 2020
 - 36 is published at ICLR 2018

Minor comments:
- Line 127: Gleaned seems an odd word to use here.
- Line 149: $\nabla_{a} \hat{f_{\Psi}}$ is repeated twice. Should one of them be $\nabla_{s} \hat{f_{\Psi}}$?
- Line 221/222: this sentence doesn't read very well.
- Line 295: should $g(x)$ be $g_{i}(x)$?
- Section 7 leads straight into 7.1
  - Consider adding an overview of Section 7 before Section 7.1 so that the reader knows what to expect.


**Questions:**

I will increase my rating if the authors address the issues raised in the Weaknesses section.

---

> ### Author Rebuttal · Authors · 2023-08-10
>
> We thank the reviewer for identifying our work's soundness and technical contributions. The valuable comments have helped us improve our manuscript. Below are our specific responses to the questions raised by the reviewer:
>
> ---
> **Weakness 1: Clarity of the figures in experiments.**
>
> - We sincerely appreciate the valuable suggestions provided by the reviewer. Due to space limitations, we chose to defer the larger-size versions of the experiment figures, namely Figure 1, 4, 6, and 7, to Appendix D.5.
> - We will make the following changes according to the reviewer's suggestions: We will modify Figure 1 by splitting it into two rows and move the hopper ablation results in Figure 4, 6, 7 to the Appendix. By implementing these changes, each row within a figure will contain only three plots, improving the overall clarity and readability of the figures.
>
> ---
> **Weakness 2: What does the shading represent? How many seeds were used in the experiments to calculate the variance?**
>
> The shaded regions in the figures correspond to standard deviation using $6$ different random seeds. We will make this clear in our paper.
>
> ---
> **Weakness 3: In Figure 7 the authors refer to the third/last columns. Consider labelling the columns a-f.**
>
> We thank the reviewer for the suggestion and will revise our manuscript accordingly.
>
> ---
> **Weakness 4: What hyperparameters were used?**
>
> The hyperparameters used in our experiments are listed in the following table, which we will also add  to our manuscript.
>
> ---
> | Hyper-parameter | Optimizer | Actor learning rate | Critic learning rate | Model learning rate | Reward learning rate | Replay buffer capacity | Mini-batch size | Discount factor | Target update rate | Activation function | Actor hidden dim | Actor hidden depth | Critic hidden dim | Critic hidden depth | Model hidden dim | Model hidden depth | Reward hidden dim | Reward hidden depth |
> |----------------|-----------|---------------------|----------------------|---------------------|----------------------|------------------------|-----------------|-----------------|--------------------|---------------------|------------------|--------------------|-------------------|---------------------|------------------|--------------------|-------------------|---------------------|
> |          | Adam      | 1e-4                | 1e-4                 | 1e-3                | 1e-3                 | 1e6                    | 512             | 0.99            | 5e-3               | ReLU                | 512              | 4                  | 512               | 4                   | 256              | 5                  | 512               | 2                   |
>
> ---
> **Weakness 5: Some citations use arXiv versions instead of conference publications.**
>
> We will fix this in our manuscript.
>
> ---
> **Minor 1: Line 127: Gleaned seems an odd word to use here.**
>
> We will change "gleaned" to "unrolled".
>
> ---
> **Minor 2: Line 149: $\nabla_a \hat{f}_\psi$ is repeated twice.**
>
> The reviewer is correct that latter one of them should be $\nabla_s \hat{f}_\psi$. We thank the reviewer for pointing this out and will fix it in our manuscript.
>
> ---
> **Minor 3: Line 221/222: this sentence doesn't read very well.**
>
> We will revise the text in Line 221/222 to "Proposition 5.2 reveals how the convergence rate changes with the variance and bias of the gradient estimators."
>
> ---
> **Minor 4: Line 295: should $g(x)$ be $g_i(x)$?**
>
> Yes. We thank the reviewer for pointing this out and will fix it in our manuscript.
>
> ---
> **Minor 5: Consider adding an overview of Section 7 before Section 7.1 so that the reader knows what to expect.**
>
> We appreciate the valuable suggestion from the reviewer. We will incorporate the following paragraph before Section 7.1 into our manuscript:
>
> "In this section, we provide empirical studies to support our theoretical findings. Firstly, in Section 7.1, we present a comprehensive evaluation of multiple algorithms derived from the proposed RP PGM framework. Using the Mujoco control tasks as our experimental domain, we compare these algorithms against various baselines to assess their performance. Secondly, in Section 7.2, we thoroughly examine the optimization challenges inherent in vanilla RP PGMs, which are characterized by issues such as exploding gradient variance and highly non-smooth loss landscapes. Subsequently, in Section 7.3, we demonstrate the effectiveness of employing smoothness regularization techniques to address these issues. Additionally, we conduct ablation studies to reveal the distinct roles played by the gradient variance and bias during training."
>
>
> ---
> We hope the reviewer could consider raising the score if we resolved the reviewer's concerns. We would be happy to have further discussions if the reviewer has any additional questions or comments.

---

> > ### Comment · Reviewer_iodf · 2023-08-10
> >
> > Thank you for the clear yet detailed rebuttal. I am happy that you will split the figures to have only 3 plots per row. I am generally happy with your changes and will increase my score. I would still advise the authors to increase the font size of the text in all of the figures.
> >
> > I also want to remind the authors that they can use an extra page for the camera-ready submission (I think), so perhaps some of the figures can stay in the main paper and do not need to be moved to the appendix.

---

> > > ### Author Response · Authors · 2023-08-11
> > > **Response**
> > >
> > > Dear Reviewer iodf,
> > >
> > > Thank you for taking the time to review our paper! We sincerely appreciate your feedback and are glad to hear that you will raise your score. Your suggestions are highly valuable, and we will carefully incorporate them into our revised manuscript, including the font size of the figure text and the figure layout. Your insightful comments have greatly contributed to improving the overall quality of our work.
> > >
> > > Best regards,\
> > > Authors

---

### Official Review · Reviewer_eACF · 2023-06-29

**Soundness:** 3 good
**Presentation:** 4 excellent
**Contribution:** 3 good
**Rating:** 7
**Confidence:** 3

**Summary:**

This paper studies differentiable model-based reparametrized policy gradient methods (RP-PGMs) with a particular focus on mitigating policy gradient variance and bias under longer horizon model rollouts to benefit agent learning and convergence. The paper introduces theorems on the bounds of policy gradient variance and bias in terms of the rollout horizon length and the lipschitz constant (``smoothness'') of the model and policy networks. Building upon the theory foundations, the paper then proposes to use Spectral Normalization on network parameters to minimize policy gradient variance and bias. Experiments on various control tasks demonstrate the effectiveness of the proposed approach.

**Strengths:**

- The paper is well-written. The theoretical and empirical findings are organized in a structured and smooth manner that is comfortable to read.

- The proposed spectral normalization approach to mitigate policy gradient variance and bias under longer model unroll horizons is both theoretically sound and empirically justified.

- The main theoretical results provide a solid theory underpinning for the convergence of reparametrized policy gradient methods.

**Weaknesses:**

- In this paper, authors set the overall spectral norm of networks to be 1. It would be interesting to investigate the impact on gradient variance, bias, and policy return when the network's spectral norm is set to a value lower than 1 during training. In particular, is there a tradeoff between policy return and gradient variance & bias?

- From Fig. 4, setting a longer model rollout horizon ($h=8/10/15$) seems to harm the sample complexity of policies, though they still converge to similar returns as the shorter horizon cases ($h=3$). It would be interesting to further empirically explore the impact of even longer $h$ on agent sample complexity (e.g., $h=20/50$). It would also be interesting to explore tasks where longer horizon model rollouts offer performance advantages over shorter horizon model rollouts. Investigating such scenarios would enhance the applicability and generalizability of the proposed approach.

**Questions:**

See "weaknesses".

**Limitations:**

Some further discussions on limitations could be added. For example, future works can explore tasks with higher dimensional observations (e.g., visual input), higher-dimensional action / control outputs (e.g., Humanoid), and alternative networks like CNNs.

---

> ### Author Rebuttal · Authors · 2023-08-10
>
> We thank the reviewer for identifying our work's soundness and technical contributions. The valuable comments have helped us improve our manuscript. Below are our specific responses to the questions raised by the reviewer:
>
> ---
> **Weakness 1: It would be interesting to investigate the impact on gradient variance, bias, and policy return when the network's spectral norm is set to a value lower than 1 during training. In particular, is there a tradeoff between policy return and gradient variance & bias?**
>
> - According to our theory and experiments, adjusting the spectral norm to a value of 1 effectively addresses the issue of exploding gradient variance and the associated optimization problems.
> - Enforcing the spectral norm to a smaller value has the potential to further reduce the gradient variance at the expense of the networks' approximation capacity. This tradeoff indicates that setting the spectral norm to a smaller value near $1$ has the potential to reach higher policy returns in certain tasks. For instance, as demonstrated in the **Figure 3 in the PDF**, we observe that setting the spectral norm to both $1$ and $0.8$ yields comparable results in the half-cheetah locomotion tasks. These settings outperform the vanilla RP-DP when spectral normalization is not applied, as well as RP-DP with a spectral norm of $0.5$. The latter exhibits increased gradient variance and gradient bias, respectively, contributing to the inferior performance.
> - Notably, the optimal spectral norm value can be highly task-dependent. For example, when the Lipschitz of the system dynamics is less than $1$, models with smaller spectral norms are still able to approximate the dynamics effectively. In contrast, in stiff systems, enforcing a small spectral norm may result in significant model error and gradient bias due to the decreased approximation capacity. Therefore, in most of our experimental tasks, we recommend using standard spectral normalization as it strikes a good balance between these considerations.
>
>
> ---
> **Weakness 2: It would be interesting to further empirically explore the impact of even longer $h$ on agent sample complexity (e.g., $h=20/50$). It would also be interesting to explore tasks where longer horizon model rollouts offer performance advantages over shorter horizon model rollouts.**
>
> - Based on our experimental observations, we found that utilizing longer expansion steps (e.g., $8$ in the hopper and walker2d tasks) combined with spectral normalization leads to improved performance compared to selecting shorter expansion steps (e.g., $h\leq 5$ as commonly used in previous methods [1, 2]).
> - However, setting an even larger $h$ value, such as $20/50$, can negatively impact training due to the compounding effect of model errors and the significant bias. This observation is in line with our theoretical results from Proposition 5.7, where we demonstrated that when model error is large, the optimal unroll step should decrease to rely more on the critic.
> - Addressing the compounding error and bias issue by learning more accurate multi-step models holds great potential as a promising avenue for future research. Once such models are developed, larger values of $h$ can potentially become advantageous, as the model surpasses the critic in accuracy. This assertion finds partial support in the results presented in Appendix D.4, where we demonstrated that learning more accurate models (e.g., by incorporating directional derivative error) leads to further performance improvements.
>
>
> ---
> **Limitation: Some further discussions on limitations could be added.**
>
> We thank the reviewer for the insightful suggestions. We would add the following discussions on the limitations of our work in a later version of our manuscript:
>
> While the proposed framework and analysis are applicable to general MDP settings, our current experiments do not cover control tasks with high dimensional inputs, such as visual observations. Additionally, our use of multi-layer perceptron networks as dynamics models restricts us from tackling more complex image input tasks. Exploring alternative model designs, such as CNN and latent models, would be a fascinating avenue for future research that we intend to pursue.
>
> ---
> We hope the above response resolves your questions and we would be happy to have further discussions if you have any additional questions or comments.
>
> ---
> [1] Clavera et al. ''Model-augmented actor-critic: Backpropagating through paths.''\
> [2] Amos et al. ''On the model-based stochastic value gradient for continuous reinforcement learning.''

---

> > ### Comment · Reviewer_eACF · 2023-08-13
> > **Reply**
> >
> > Thanks authors for the rebuttal! I'd like to keep by current ratings as they are already high.

---

### Official Review · Reviewer_yXDQ · 2023-07-03

**Soundness:** 4 excellent
**Presentation:** 4 excellent
**Contribution:** 3 good
**Rating:** 7
**Confidence:** 3

**Summary:**

This paper examines reparameterization policy gradient methods in model-based reinforcement learning. It investigates the relationship between the convergence rate, bias and variance of reparameterization policy-gradient, smoothness of the model, and approximation error. Based on the theoretical analysis, it further proposes a spectral normalization method to enforce smoothness on the model and policy.

**Strengths:**

1. The paper is well-written and easy to follow.
2. It is interesting to note how different parts of MB PR PGMs interact with each other, and how the smoothness of the model affects the model expansion steps.
3. The experimental results demonstrate that applying spectral normalization to regularize the model and policy leads to improved performance and enables longer model expansion.

**Weaknesses:**

1. The application of spectral normalization to deep networks is not new.
2. Applying spectral normalization limits both the model and policy capacity, which can result in an increasing gradient bias and a slower convergence rate. The trade-off between gradient variance and bias has not been thoroughly studied in experiments. (e.g. Section 7.3 only studies variance and bias on two disjoint environment suites)

**Questions:**

No

**Limitations:**

The authors have adequately addressed the limitations

---

> ### Author Rebuttal · Authors · 2023-08-10
>
> We thank the reviewer for identifying our work's soundness and technical contributions. The valuable comments have helped us improve our manuscript. Below are our specific responses to the questions raised by the reviewer:
>
> ---
> **Weakness 1: The application of spectral normalization to deep networks is not new.**
>
> - Previous studies have primarily focused on utilizing spectral normalization to tackle training instability problems associated with deep neural networks, such as [1, 2].
> - In contrast, our spectral normalization method is rooted in a theoretical analysis of model-based reparameterization policy gradient methods and aims to address the issue of exploding variance that can arise when employing lengthy model unrolls.
> - Additionally, our investigation in Appendix D.3 reveals that applying spectral normalization to other model-based reinforcement learning algorithms can yield adverse effects. This suggests that the applicability of smoothness regularization is not universal, and we specifically employ spectral normalization based on theoretical motivations, tailored for model-based reparameterization policy gradient methods.
>
> ---
> **Weakness 2: The trade-off between gradient variance and bias has not been thoroughly studied in experiments. (e.g. Section 7.3 only studies variance and bias on two disjoint environment suites).**
>
> - In our ablation studies on the gradient bias, we demonstrated a consistent result: even in cases where the bias is minimal (e.g., RP-DP without spectral normalization) or completely absent (e.g., analytic policy gradient approach), the amplified variance can still lead to poor performance outcomes.
> - The task designs and agent configurations are the same in the Mujoco and dFlex environment. We opted to use the *differentiable* dFlex simulator in order to implement the analytic policy gradient approach. This choice also enhances the precision and computational efficiency of our analysis as the bias can be calculated by directly comparing the model-based RP gradient and the analytic gradient.
>
> ---
> We hope the above response resolves your questions and we would be happy to have further discussions if you have any additional questions or comments.
>
> ---
> [1] Miyato et al. ''Spectral normalization for generative adversarial networks.''\
> [2] Bjorck et al. ''Towards deeper deep reinforcement learning.''

---

> > ### Author Response · Authors · 2023-08-16
> >
> > Dear Reviewer yXDQ,
> >
> > As we are approaching the midpoint of the discussion period, we would like to cordially inquire about the extent to which we have successfully addressed the concerns outlined in your review. Should there be any lingering points that require further attention, please rest assured that we are enthusiastic about the opportunity to provide comprehensive responses to any subsequent queries or comments you may have.
> >
> > Your constructive input remains invaluable to us, and we appreciate your dedication to enhancing the quality of our manuscript. Thank you for your time and consideration.
> >
> > Best,
> >
> > Authors

---

### Official Review · Reviewer_epDC · 2023-07-06

**Soundness:** 3 good
**Presentation:** 3 good
**Contribution:** 2 fair
**Rating:** 6
**Confidence:** 4

**Summary:**

The paper theoretically analyzes the reparameterization policy gradient estimator’s bias and variance in reinforcement learning optimization and provide results characterizing the optimization convergence using such gradient estimators. It then proposes to apply spectral normalization (dividing the linear weight matrix by its largest singular value to make the Lipschitz constant upper bounded by $1$) on the learned world model and learned policy network. Empirically, they observe the spectral normalization improves the variance of the RP gradient estimator and improves the performance of RP to be comparable/better than other RL methods (including likelihood ratio methods).

**Strengths:**

1. The paper does a good job setting up the background and context about the policy gradient estimation methods in reinforcement learning.

2. The experimental results confirm the benefits of applying spectral normalization in reparameterization policy gradient methods when the model expansion steps are large.

**Weaknesses:**

1. Despite having an extensive background discussion on policy gradient, the paper is very brief on describing the formula and implementation specifics of model derivatives on predictions (DP) and model derivatives on Real Samples (DR): Equation (4.1) and (4.2) are plain tautology (RHS only expands the reward function $J$) and provides no information the backward recursive structure of the gradient estimators.

2. Assumption 5.3 on Lipschitz Continuity. In the paper, the authors assume a Lipschitz constant on the learned world model $f_\psi$. However, this neglects the fact that the learned world model is also changing as the learning progresses. Without spectral normalization of the world model, it is conceivable that the world model might have a growing Lipschitz constant over the update steps $T$. It’s not clear to me that this assumption has already taken into account of this update time aspect and deserves further explanation in the paper.

3. The paper conducts experiments on Mujoco tasks.  To my understanding, Mujoco tasks only have randomness in the initialization state, but doesn’t have randomness in the state transition. As a result, the assumptions on stochastic transitioning environment made in the paper don’t seem to hold. It would be necessary for the authors to clarify whether this is the case or why they haven’t experimented with environments with greater randomness (or make Mujoco random).

4. Convergence theory is non-informative.
    - In proposition 5.2, to have the learned model converge to a stationary point, we want the LHS (minimum gradient’s squared 2-norm encountered so far) to be decreasing as a function of $T$. However, in the upper bound on the right hand side, the first term contains $\mathbf{E}[J(\pi_{\theta_T}) - J(\pi_{\theta_1})] $ which should increase as $T$ increases if the optimization is making progress in maximizing the value function. The relationship between this term and its denominator $T$ should be further discussed. Besides, the second term on the right hand side has a term $O(\frac{\sum_{t=0}^{T-1} v_t}{T})$. In the Proposition 5.4, the authors provide a $O(1)$ bound for $v_t$ for a fixed number of gradient estimates $N$. Thus the term $O(\frac{\sum_{t=0}^{T-1} v_t}{T})$ would only be $O(1)$. Hence it’s not clear to me this theory can capture the empirical observation that training using reparameterization gradient can converge to approximate local (or even global) maxima (which are stationary points).
    - In Proposition 5.7, the author gives a big O notation for the optimal model expansion step $h^*$ for update iteration $t$. This bound depends on the gradient errors $\epsilon_{v, t}$ and $\epsilon_{f, t}$, both of which are unknown values that depend on the ground truth value function and world model. As a result, it’s not clear whether this theory can offer any practical guidance. Besides, this optimal expansion step is update-iteration dependent (depends on $t$) and it’s not clear how to practically instantiate such an h-schedule.

**Questions:**

1. **Correlated randomness over time** Line 129 and 138 are reusing the same random variable $\zeta$ and $\xi$ for all time steps’ action sampling and environment transition in the same rollout. However, the randomness in action sampling and the randomness in state transition shouldn’t be correlated over different time steps. Can the authors clarify why these variables are shared for a given rollout trajectory?

2. **Are there other ways to trade off bias for variance?** The authors propose to use spectral normalization to reduce the variance at the cost of bias. Another way to potentially trade off bias for variance is to use something similar to truncated back propagation through time. In this case, one could imagine using a very small time horizon and completely ignore the discounted rewards after a certain time step (currently captured by the learned critic function). How would such a (high bias, low variance) remedy perform in comparison to the SN methods on the RL tasks considered in this paper?

3. **Would there be sufficient incentive to use longer expansion step + spectral normalization?** Looking at the experiment figures 4, 5, and 6, it seems that the performance of RP gradient only starts to deteriorate for longer unroll length (h > 5). When h ≤ 5, it seems that using SN doesn’t really improve the performance. In this case, why wouldn’t researchers just choose to use a short model expansion step (together with a learned critic) and not to use the spectral normalization approach proposed by the authors?

**Limitations:**

The paper doesn’t have much discussion on the limitations. I would encourage the authors discuss what they think are the limitations in their theoretical and experimental results. I don’t think negative societal impacts are relevant to this paper.

---

> ### Author Rebuttal · Authors · 2023-08-10
>
> We thank the reviewer for the valuable comments. Below are our specific responses to the questions raised by the reviewer:
>
> **Weakness 1: Eq. 4.1 and 4.2 are plain tautology and provides no information the backward recursive structure of the gradient estimators.**
>
> - Eq. 4.1 and 4.2 (or A.9) depict the two variants of RP gradients where the sole distinction lies in the incorporated noise variables $\zeta$ and $\xi$, which are sampled from predefined distributions and inferred from real samples, respectively.
> - These equations first compute the value and then obtain its first-order gradient w.r.t. $\theta$, whose recursive expression is given in Appendix A. In practice, the gradient is computed using modern deep learning libraries by automatic differentiation.
>
>  **Weakness 2:  The model might have a large Lipschitz constant over the update steps.**
>
> - The Lipschitz constant of the *learned* model is a global coefficient that can be enforced by constraints or model designs like SN.
> - In fact, what we aim to address is the case when the model has a large Lipschitz constant that can result in significant variance in gradients. In such cases, it becomes crucial to employ smoothness regularization techniques to effectively tackle the optimization challenges that arise.
>
> **Weakness 3: Randomness in the state transition.**
>
> - Our results hold for MDPs with both stochastic and deterministic transitions. For the latter case, the noise $\xi^*$ would be $0$ for the dynamics $s'=f(s, a, \xi^*)$. However, the issue of exploding variance still exists due to the randomness in the initial state and the stochastic policy.
> - In **Figure 1 in the PDF**, we report the results in Mujoco tasks with stochastic state transitions.
>
> **Weakness 4.1: The relationship between $\mathbb{E}[J(\pi_{\theta_T}) - J(\pi_{\theta_1})]$ and $T$ in Proposition 5.2. Besides, the RHS has a variance term that would be $O(1)$.**
>
> - For MDPs with bounded reward, e.g., if $|r(s, a)| \leq r_m$ as stated in Proposition 5.2, then $\mathbb{E}[J(\pi_{\theta_T}) - J(\pi_{\theta_1})]\leq 2r_m$, which is a universal upper bound. This is because $|V^\pi(s_0)|=(1-\gamma)|\mathbb{E}_\pi[\sum_i\gamma^i r(s_i, a_i)]|\leq r_m$.
> - Proposition 5.4 shows that $v_t$ scales as $O(1/N)$. It suffices to choose a reasonably large value for $N$ for convergence, as evidenced by Corollary 5.9. Even so, we emphasize that Proposition 5.2 primarily serves to justify our SN method by characterizing the roles of gradient bias and variance in convergence. To serve this purpose, we explicitly express the bias and variance terms while imposing minimal assumptions. As a result, the upper bound in Proposition 5.2 may appear looser compared to the typical bounds in model-free RL analysis, which involve fewer error sources and stronger assumptions (e.g., the bounded variance Assumption 4.4 in [1].).
>
>
> **Weakness 4.2: Optimal model expansion step in practice.**
>
> Proposition 5.7 aims to shed light on the factors influencing $h^*$ and its dependence on the horizon and model, critic errors. The result indicates that $h^*$ should increase when the model is more accurate and decrease when the critic is more accurate. However, this is only a rough guidance in practice since accurately quantifying these errors poses a significant challenge, making it difficult to determine an optimal $h^*$ schedule during training.
>
>
> **Question 1: Correlated randomness over time.**
>
> At each timestep $i$, the random variables $\zeta$ and $\xi$ are independently sampled and are not shared within a rollout trajectory. We will revise the notations in Line 129 to $\zeta_i, \xi_i$, and in Line 138 to  $\zeta_{i,n}, \xi_{i,n}$. We sincerely appreciate the reviewer for bringing this to our attention.
>
>
> **Question 2: Are there other ways to trade off bias for variance?**
>
> - The reviewer is correct that one way to reduce gradient variance is to use Truncated BPTT. However, this approach can lead to a huge gradient bias, as it over-prioritizes short-term dependencies. This has been observed in some previous works [2].
> - In order to further investigate this, we performed additional experiments and report the performance results in the **Figure 2 in the PDF**. These experiments demonstrate that Truncated BPTT struggles to achieve high returns.
>
>
> **Question 3: Would there be sufficient incentive to use longer expansion step + SN?**
>
> - Our experimental results demonstrated that longer expansion steps (e.g., $h=8$ in hopper and walker2d) with SN offers a better performance than choosing a short expansion steps (e.g., $h\leq 5$ as commonly used in previous methods).
> - The observed decrease in performance with larger $h$ is in line with our theoretical findings from Proposition 5.7 as the compounding model error becomes more pronounced. Therefore, it is advisable to decrease $h$ and rely more on the critic. Notably, SN is proposed to mitigate the optimization challenges, applying which the above tradeoff between model and critic error is valid.
> - Learning more accurate models to address the bias issue holds significant promise for future research. Once we develop such models, larger $h$ can become advantageous, as the model surpasses the critic in accuracy.
>
> **Limitation: The paper doesn’t have much discussion on the limitations.**
>
> While SN addresses the challenge of exploding gradient variance and enables longer model unrolls, learning more accurate longer-horizon models to fully exploit the gradient information remains an open problem. Besides, our experiments focus solely on SN as a smoothness regularization technique, and we acknowledge the need for further exploration of alternative designs.
>
> We hope the above response resolves your questions and would be happy to have further discussions if you have any additional comments.
>
> [1] Wang et al. ''NPG methods: Global optimality and rates of convergence.''\
> [2] Xu et al. ''Accelerated policy learning with parallel differentiable simulation.''

---

> > ### Author Response · Authors · 2023-08-16
> > **Follow-up on the rebuttal.**
> >
> > Dear Reviewer epDC,
> >
> > As we are approaching the midpoint of the discussion period, we would like to cordially inquire about the extent to which we have successfully addressed the concerns outlined in your review. Should there be any lingering points that require further attention, please rest assured that we are enthusiastic about the opportunity to provide comprehensive responses to any subsequent queries or comments you may have.
> >
> > Your constructive input remains invaluable to us, and we appreciate your dedication to enhancing the quality of our manuscript. Thank you for your time and consideration.
> >
> > Best,
> >
> > Authors

---

> > > ### Comment · Reviewer_epDC · 2023-08-17
> > > **Response to Authors' Rebuttal**
> > >
> > > I would like to thank the authors for their rebuttal and additional experiment results. The authors have satisfactorily answered my identified Weakness 2, Weakness 4, Question 1, Question 2. For the rest of my comments,
> > >
> > > - __Weakness 1__: I understand that DP and DR are two different ways to compute the reparameterization gradient through the learned model. However, I think these two equations (4.1) and (4.2) do not point to these differences. Instead it’s the sentences that follow these equations (Line 138 and Line 153-154) that describe these differences (whether using model predictions or real samples). Thus my view that these two equations do not provide much information (thus are more tautology) still remains unchanged.
> > > - __Weakness 3__: The authors have acknowledged that their original experiments on Mujoco indeed haven’t captured the stochastic transition dynamics and have provided new experiments by adding Gaussian noise to the state transitions to the default Mujoco transitions. I think this is a good first step, but I would still recommend the authors to consider other possible ways of inducing transition randomness (for example, adding randomness to the applied action instead of the transitioned states so that the randomness goes through nonlinearity) or other RL experiments that exhibit true transition randomness. Besides, I would strongly recommend the authors to include the discussion about these stochastic results more explicitly in the main paper to make the evaluation complete.
> > > - __Question 3__: The authors claim that longer expansion steps with SN offers better performances ($h=8$ on hopper and walker2d). I’m not sure if the authors are referring to Figure 5. If so, I would like to point out that
> > > 1. It seems to me that the confidence intervals of the rewards of $h=8$ might be intersecting with other smaller expansion steps’ confidence intervals, so it’s not clear to me whether $h=8$ is indeed better in terms of the total rewards.
> > >
> > > 2. It is also important to consider the sample complexity (number of episodes) needed to reach a given reward threshold when comparing different expansion step lengths. My concern is that under this sample complexity measure, it might still be more beneficial to use shorter expansion step than longer ones. I would appreciate having the authors respond to these two comments.
> > >
> > > - __Limitations__: The authors mention that “learning more accurate longer-horizon model” as an open problem. I wonder whether the authors believe having a more accurate model would be inherently contradictory to the type of spectral normalization techniques the authors propose.

---

> > > > ### Author Response · Authors · 2023-08-17
> > > > **Response to Reviewer epDC**
> > > >
> > > > We appreciate the reviewer's valuable feedback. Below are our responses to the raised questions:
> > > >
> > > > **Weakness 1: Eq. (4.1) and (4.2) do not point to the differences between DP and DR. Instead it’s the sentences that follow these equations that describe these differences.**
> > > >
> > > > We will make the following modifications to enhance the clarity:\
> > > > Firstly, we'll introduce the unified form of the RP gradient at the beginning of Sec. 4.2:
> > > > $\hat{\nabla}\_\theta J(\pi\_{\theta}) = \frac{1}{N}\sum\_{n=1}^N \hat{\nabla}\_\theta V^{\pi\_\theta}(\hat{s}\_{0,n}) = \frac{1}{N}\sum\_{n=1}^N \nabla\_\theta(\sum\_{i=0}^{h-1} \gamma^i r(\hat{s}\_{i,n}, \hat{a}\_{i,n}) +\gamma^h \hat{Q}\_\omega(\hat{s}\_{h,n}, \hat{a}\_{h,n})),$
> > > > where $\hat{s}\_{0,n}\sim\mu\_{\pi\_\theta}$, $\hat{a}\_{i,n}=\pi\_\theta(\hat{s}\_{i,n}, \varsigma\_{i,n})$, and $\hat{s}\_{i+1,n} = \hat{f}\_\psi(\hat{s}\_{i,n}, \hat{a}\_{i,n}, \xi\_{i,n})$.\
> > > > Next, we'll describe how the DP and DR gradients are estimated by defining the corresponding noises $\varsigma\_{i,n}$ and $\xi\_{i,n}$. We will: (i) substitute Eq. (4.1) with $\varsigma\_{i,n}\sim p(\varsigma), \xi\_{i,n}\sim p(\xi)$ to depict RP-DP, and (ii) describe RP-DR by replacing Eq. (4.2) with " $\varsigma\_{i,n}$ and $\xi\_{i,n}$ are inferred by solving $a\_{i,n} = \pi\_\theta(s\_{i,n}, \varsigma\_{i,n}), s\_{i+1,n} = \hat{f}\_\psi(s\_{i}, a\_{i,n}, \xi\_{i,n})$ to parameterize the real data sample $(s\_{i,n}, a\_{i,n}, s\_{i+1,n})$". We'll also include further specifications: "Take 1d Gaussian models $\hat{s}\_{i+1}\sim\mathcal{N}(\phi(\hat{s}\_{i}, \hat{a}\_{i}), \sigma^2)$ as an example, where the mean is the output of a function $\phi$. Then $\xi\_{i,n} = (s\_{i+1,n} - \phi(s\_{i,n}, a\_{i,n})) / \sigma$".
> > > >
> > > > **Weakness 3: Other ways of inducing transition randomness (e.g., adding randomness to the applied action so that it goes through nonlinearity) or other RL experiments. Besides, I recommend the authors to include the discussion about these stochastic results in the main paper.**
> > > >
> > > > - We'd like to emphasize that the results in this paper are applicable to both stochastic and deterministic transitions. By ''adding randomness to the applied action'', does the reviewer mean stochastic policies such as Gaussian policies employed in our experiments? In these policies, Gaussian noises are added to the action means. Using stochastic rather than deterministic policies introduces randomness into the state transitions in a nonlinear way even for deterministic transitions.
> > > > - We appreciate the reviewer's suggestion, and we will incorporate the additional stochastic results with discussions into the main paper.
> > > >
> > > > **Question 3.1: The reward confidence intervals of $h=8$ might be intersecting with other smaller expansion steps, so it’s not clear to me whether it is indeed better.**
> > > >
> > > > - The primary goal of Fig. 2-6 is to demonstrate that applying SN can alleviate the exploding variance issue, enabling longer model unrolls without significant optimization difficulties. *However, a larger $h$ is not necessarily better*. As shown by Proposition 5.7, there is a tradeoff between model error and critic error that determines an optimal $h^*$. This result holds when SN is applied.
> > > > - The limited gains when increasing $h$ to $8$ align with this tradeoff. A large $h$ may benefit training when the critic is inaccurate. Meanwhile, increasing $h$ can be detrimental due to the compounding effect of model errors. Determining the optimal $h^*$ is challenging as it depends on the model and critic error.
> > > > - As shown in Appendix D.4, learning more accurate models (by incorporating directional derivative error) leads to further performance improvements. This serves as evidence supporting that the constrained improvement partly attributes to inaccurate models.
> > > >
> > > > **Question 3.2: It is also important to consider the sample complexity needed.**
> > > >
> > > > The reported returns were attained by employing an equal number of samples for various $h$ in each task (1e6 for walker2d and 4e5 for hopper). We didn't observe obvious distinctions in the needed samples for different $h$ when SN is applied.
> > > >
> > > > **Limitations: I wonder whether the authors believe having a more accurate model would be inherently contradictory to SN.**
> > > >
> > > > The primary objective of the SN method is to tackle the variance issue and the associated optimization challenges. By employing SN, we can navigate the tradeoff between model error and critic error, encouraging us to learn more accurate models and effectively utilize the model gradient information. This is especially true for non-stiff dynamics with small Lipschitz constants, in which case the models with SN still reside within the function class of the true dynamics. For stiff systems, a balance should be struck between the gradient variance and bias, where applying SN to an accurate (thus non-smooth) model reduces the variance with the cost of increased bias. For a more comprehensive discussion, please refer to Line 50-56 and 242-247.

---

> > > > > ### Comment · Reviewer_epDC · 2023-08-17
> > > > > **Follow up to Authors' Response**
> > > > >
> > > > > I would like to thank the authors for their prompt response. My questions have been (partially) answered. Given the overall answers in the two responses, I will increase my score to 6. I make comments about each individual point below:
> > > > >
> > > > > **Weakness 1**: What the authors propose as a modification to the introduction of DP and DR has resolved the tautology problem I initially described.
> > > > >
> > > > > **********Weakness 3**********: I acknowledge that the authors have been using stochastic policies that already adds randomness to the action, which might make my suggestion of adding additional noise to create stochastic transitions unnecessary, but I still believe exploring other environments with stochastic transitions could improve the comprehensiveness of the experiments.
> > > > >
> > > > > ************************Question 3************************: The reason I ask whether $h=8$ is the best step size stems from my original question that why we should even consider using spectral normalization when we can already have good performance without SN by just having a small $h$ value (e.g. $h=3$). I don’t believe the authors’ two responses have resolved this concern and the authors can further clarify this point if they have time.
> > > > >
> > > > > ********************Limitation********************: The authors mention “applying SN to an accurate model”. This language makes it sound like SN is applied after an accurate (non-smooth) model has already been trained. However, I am under the impression that the authors have been applying SN as part of the model even during its training. If this is true, under this model Lipschitzness constraint (enforced during training), I’m doubtful how one can obtain a much more accurate environment model in the first place.

---

> > > > > > ### Author Response · Authors · 2023-08-18
> > > > > > **Response to Reviewer epDC**
> > > > > >
> > > > > > We appreciate the reviewer for the valuable feedback. Below are our specific responses to each of the comments:
> > > > > >
> > > > > > **Weakness 1:** We are glad to know that our proposed modification has addressed this issue.
> > > > > >
> > > > > > **Weakness 3:** We are delighted that our adoption of stochastic policies has addressed the reviewer's concern. We agree that exploring other environments with stochastic transitions beyond Mujoco will improve the comprehensiveness of our experiments, which we leave as future work.
> > > > > >
> > > > > > **Question 3:** In our experiments, we observed that the application of spectral normalization (SN) yields consistent improvements over the vanilla counterpart without SN. As shown in Figure 1, RP-DP-SN outperforms both the vanilla RP-DP and other baselines across nearly all evaluated tasks, with particularly significant enhancements in tasks such as ant locomotion. Therefore, there *is* sufficient incentive for the adoption of SN. Furthermore, our empirical study on the model expansion steps $h$ primarily aims to demonstrate that employing SN effectively mitigates the issue of exploding variance and enables longer model unrolls. The optimal choice of $h$ and the specific improvement in each environment greatly depend on the task at hand and the tradeoff between model error and critic error. Thus, determining whether a larger value of $h$ is beneficial requires careful consideration of the task's characteristics.
> > > > > >
> > > > > > **Limitation:** In scenarios where the system dynamics exhibit non-stiff behavior, characterized by small Lipschitz constants, the application of SN does not compromise the approximation capacity of the model. In fact, the model remains within the function class that adequately represents the dynamics. On the other hand, when the system dynamics are stiff, SN may indeed decrease the model's approximation capacity. In such cases, the reviewer is correct that obtaining an accurate model becomes challenging. However, even in the presence of SN, we are still able to incorporate additional techniques to enhance the accuracy of longer-horizon model predictions, which is the open problem that we described.
> > > > > >
> > > > > > We will incorporate the above discussions into our paper. We sincerely thank the reviewer for the valuable feedback and insightful suggestions, which have greatly helped us improve our manuscript.

---

### Author Rebuttal · Authors · 2023-08-10

We conduct additional experiments to address Weakness 3 and Question 2 raised by **Reviewer epDC**, and Weakness 1 raised by **Reviewer eACF**. The results can be found in the attached PDF file.

---

### Decision · Program_Chairs · 2023-09-21

**Decision:**

Accept (poster)

**Comment:**

This paper had strong support from a majority of its reviewers. The reviewers appreciated the theoretical and empirical justifications of the spectral normalization scheme in confining bias on long unrolls of the model, unified synthesis of model-based and policy gradient methods (highlighting the role of smoothness) and introduction of a practical algorithm. Therefore, we are happy to recommend the paper for acceptance.